# The Cytnx Library for Tensor Networks

Kai-Hsin Wu[1,*], Chang-Teng Lin [2,3], Ke Hsu[2,3], Hao-Ti Hung[2,3], Manuel Schneider [4], Chia-Min Chung [5,6,7], Ying-Jer Kao [2,3,8†], Pochung Chen [6,9,10*]

**1** Department of Physics, Boston University, Boston 02215, United States of America
**2** Department of Physics, National Taiwan University, Taipei 10607, Taiwan
**3** Center for Theoretical Physics, National Taiwan University, Taipei 10607, Taiwan
**4** Institute of Physics, National Yang Ming Chiao Tung University, Hsinchu 30010, Taiwan
**5** Department of Physics, National Sun Yat-Sen University, Kaohsiung 80424, Taiwan
**6** Physics Division, National Center for Theoretical Sciences, Taipei 10617, Taiwan
**7** Center for Theoretical and Computational Physics, National Sun Yat-Sen University, Kaohsiung 80424, Taiwan
**8** Center for Quantum Science and Technology, National Taiwan University, Taipei 10607, Taiwan
**9** Department of Physics, National Tsing Hua University, Hsinchu 30013, Taiwan
**10** Frontier Center for Theory and Computation, National Tsing Hua University, Hsinchu 30013, Taiwan
† yjkao@phys.ntu.edu.tw ⋆ pcchen@phys.nthu.edu.tw * kaihsinwu@gmail.com

January 5, 2024

## Abstract

We introduce a tensor network library designed for classical and quantum physics simulations called Cytnx (pronounced as *sci-tens*). This library provides almost an identical interface and syntax for both C++ and Python, allowing users to effortlessly switch between two languages. Aiming at a quick learning process for new users of tensor network algorithms, the interfaces resemble the popular Python scientific libraries like NumPy, Scipy, and PyTorch. Not only multiple global Abelian symmetries can be easily defined and implemented, Cytnx also provides a new tool called `Network` that allows users to store large tensor networks and perform tensor network contractions in an optimal order automatically. With the integration of cuQuantum, tensor calculations can also be executed efficiently on GPUs. We present benchmark results for tensor operations on both devices, CPU and GPU. We also discuss features and higher-level interfaces to be added in the future.

# 1 Introduction

Tensor networks have been well established as an important tool in various applications including, but not limited to, many-body systems in condensed matter physics [1–8] and for quantum chemistry [9–15]. In high energy physics, tensor networks can provide an alternative where established Monte Carlo methods can not be used [16–19]. Recently, tensor network methods have also been applied to machine learning [20–23], computational fluid dynamics [24, 25] and have emerged as an efficient method to simulate quantum circuits [26–28]. Therefore, having a tensor library is crucial to accelerate the development of tensor network algorithms. With such a tool, users can focus on the applications and use state-of-the-art, pre-tested, and optimized tensor operations provided by the library, instead of investing time in implementing and testing basic tensor functionalities. Here, we present a new open-source tensor network library – Cytnx [29, 30] (pronounce as *sci-tens*)[1], which provides an easy-to-use framework for beginners with extensive and powerful functionality for advanced tensor network applications [29].

## 1.1 Overview of the Cytnx Design

There exist several tensor libraries, each addressing various aspects of the tensor computation [31–37]. In Cytnx, we envision a framework designed and optimized for higher dimensional tensor networks, such as projected entangled pair states (PEPS) or multi-scale entanglement renormalization ansatz (MERA). Compared to the matrix product state (MPS), these tensor networks have higher connectivity, which makes keeping track of tensor indices and connectivity cumbersome. Therefore, we design a `Network` class to store the blueprint of a tensor network. This enables under-the-hood optimizations for tensor contractions. Moreover, the `Network` can be reused for further contractions by loading a new set of tensors.

On the one hand, we make Cytnx easy to interface with Python packages such as NumPy and Scipy and machine learning frameworks such as PyTorch. This allows Cytnx users to take advantage of the available Python libraries supported by the community. Moreover, new users can easily port their existing Python code to take advantage of the high-level design of Cytnx.

On the other hand, for the performance minded users, Cytnx keeps most of the Application Programming Interfaces (APIs) exactly the same for both C++ and Python by design; therefore, users can quickly prototype their algorithms in Python, and switch to C++ with minimal rewriting of the code base for further optimization. In Cytnx, unlike in many libraries where the Python interface serves as a wrapper, both C++ and Python interfaces have direct access to the underlying tensor containers, giving users greater flexibility.

Cytnx also supports GPU computing. When creating a tensor, Cytnx allows users to specify the device for storing the data. Similar to PyTorch, users can use the same API calls regardless of the devices where the tensors are kept. This allows users to benefit from the GPU acceleration with minimal changes for heterogeneous computing environments. In addition to the traditional CUDA libraries, Cytnx also incorporates functions from cuQuantum, allowing more efficient usage of GPU when performing tensor contractions.

---

[1]The name Cytnx is derived from the following keywords, C++(C), Python(y), and Tensor Networks (tnx).

Another easy-to-use feature of Cytnx is the ability to access indices of tensors by user-defined labels. Once meaningful index labels are set, the ordering and permutation of the indices in tensors can be handled by the library automatically. This way, users can focus on their application instead of dealing with the technical details of the tensors and how the information is stored.

Moreover, Abelian symmetries and quantum numbers can be easily included in tensors. With symmetric tensors, Cytnx ensures that all operations respect the symmetries of the system, and makes use of an efficient block structure of the tensors. Directional bonds are required for symmetric tensors, but can also be defined without being linked to symmetries. Cytnx ensures that only bonds with opposite directions can be contracted, which makes the code more resilient against errors.

## 1.2 How to use this document

This document aims to give an overview of the functionality and usage of Cytnx. Users can find more information about the library in the User Guide [30]. Also, an API documentation of the classes and functions in Cytnx is available online [38].

Here we present example codes in Python, with some comments on C++ in Sec. 3. Advanced users can refer to the online documentation [30] for C++ example codes. All code examples in this document assume a successful installation of Cytnx. In order to run the Python examples, the user needs to insert this line before the example code:

```
import cytnx
```

Building from the source code and Conda installation of Cytnx are both available. See the online documentation for detailed installation instructions [30]. The explanations here refer to version 0.9.7 of the Cytnx library.

This article is organized as follows. In Sec. 2 we introduce the graphical tensor notation and provide a simple example showing how to use Cytnx to implement a tensor contraction. Section 3 explains basic concepts and general naming conventions of the library. Sections 4 to 6 show the creation and manipulation of tensors and their indices, where Sec. 6 introduces symmetries and symmetric tensors. Tensor contractions are covered in detail in Sections 7 and 8. In Sec. 9 we explain how to decompose tensors and use linear algebra functions. The performance of Cytnx is benchmarked against ITensor [31] in Sec. 11. Finally, we give a summary and discuss the future road map of Cytnx in Sec. 12.

## 2 Sneak preview: tensors and tensor contraction

In this sneak preview, we start from the standard mathematical symbolic notation of a tensor and introduce the graphical notation. With this, we show how the tensor contraction can be represented transparently. Finally, we demonstrate how to convert the graphical notation into corresponding code in Cytnx and perform tensor contractions.

A graphical notation for tensors and tensor networks is very useful to visualize the tensor and the connectivity in tensor network algorithms. We introduce our graphical convention in the following. A tensor can be thought of as a multi-dimensional array. For example, a rank-0 tensor is a scalar, a rank-1 tensor is a vector and a rank-2 tensor is a matrix. Mathematically, a rank-$N$ tensor can be written in symbolic notation as:

$$T_{i_1, i_2 \cdots, i_N},\tag{1}$$

where $i_1, i_2, \cdots, i_N$ are the indices of the tensor. This notation is standard, but for a tensor

network that involves a series of tensor contractions, the expression soon becomes complicated and difficult to read. This is where the graphical tensor notation is useful.

In the graphical notation, each tensor is represented by a node (sometimes it is also called a vertex in graph theory) and several bonds (legs) attached to it. Each bond corresponds to an index of the tensor. The number of bonds equals the rank of a tensor. Fig. 1 shows examples of several tensors.

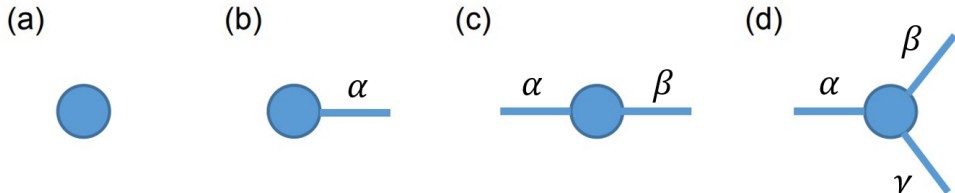

Figure 1: Graphical notation of tensors. (a) scalar; (b) vector; (c) matrix; (d) rank-3 tensor.

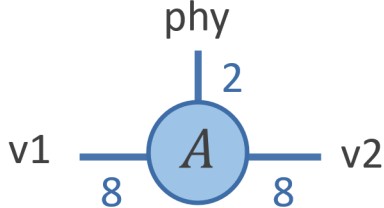

Figure 2: Visualization of the tensor constructed in Listing 1, with the dimension and label of each legs.

Consider a rank-3 tensor A as shown in Fig. 2 with indices labeled by "v1","phys","v2" and dimensions $8, 2, 8$ respectively. The following code example (Listing 1) shows how to create a `UniTensor` to represent such a tensor and initialize all the tensor elements to one.

```
A = cytnx.UniTensor.ones([8,2,8]).relabels(["v1","phy","v2"])\
                     .set_name("A")
# or, using argument names:
A = cytnx.UniTensor.ones([8,2,8], labels=["v1","phy","v2"], name="A")
```

Listing 1: Create a rank-3 tensor. Two styles are shown here. The first syntax is compatibility with C++ and is recommended. In the second syntax we use keyword arguments in Python.

One of the most important operations in tensor network algorithms is the tensor contraction. It is an operation that sums the product of tensor elements over some common indices. For example, consider the contraction of three tensors $A$, $B$, and $C$:

$$D_{\alpha\delta} = \sum_{\beta\gamma} A_{\alpha\beta\gamma} B_{\beta\delta} C_{\gamma} \tag{2}$$

In this example, $A$ has three indices labeled by $\alpha$, $\beta$, and $\gamma$. $B$ has two indices labeled by $\beta$ and $\delta$, and $C$ has one index labeled by $\gamma$. Equation (2) defines a tensor contraction that sums over

the common indices $\beta$ and $\gamma$, and the result is a tensor $D$ with two indices labeled by $\alpha$ and $\delta$. The corresponding graphical notation is shown in Fig. 3. Indices that connect two nodes are assumed to be summed over. The dashed lines indicate the connection of two bonds with the same indices to be summed.

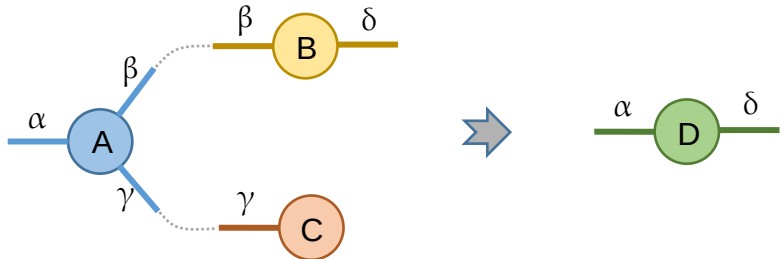

Figure 3: Graphical notation for the tensor contraction in Eq. (2)

Cytnx provides several methods for tensor contractions. One can use `Contract` or `Contracts` to perform simple contraction of two or more tensors respectively. They sum over common labels on these tensors. These functions are easy to use and make the code readable. The following code example (Listing 2) demonstrates the contraction of three tensors `A`, `B` and `C`. We note in passing that the user needs to take care of the correct index labeling in advance. See Sections 7.1 and 7.2 for details.

```python
# Initialize UniTensors and their labels
A = cytnx.UniTensor.ones([2,2,2])\
                    .set_name("A").relabels(["alpha","beta","gamma"])
B = cytnx.UniTensor.ones([2,2])\
                    .set_name("B").relabels(["beta","delta"])
C = cytnx.UniTensor.ones([2])\
                    .set_name("C").relabels(["gamma"])
# Do the contractions
AB = cytnx.Contract(A, B)
D = cytnx.Contract(AB, C)
# Similarly, in one step:
D = cytnx.Contracts([A,B,C])
```

Listing 2: Using `Contract` or `Contracts` to perform tensor contractions.

In a realistic algorithm, the number of tensors in a tensor network contraction can be quite many. The overall contraction is typically implemented in a sequence of binary contraction of tensors. However, the order of these binary contractions is not unique, and the computational costs depend strongly on the contraction order. One typically needs to first find an optimal contraction order before implementing the contractions.

In Cytnx, we introduce a powerful `Network` contraction method, which allows one to define a tensor contraction in an abstract way. The actual tensors to be contracted do not need to have a certain index order or specific labels. Consequently, the same `Network` object can be reused for successive contractions. Moreover, the contraction order can be optimized automatically. When a `Network` contraction is launched, it computes an optimized contraction order by a Dynamic Programming Algorithm. The optimized order can be saved and re-used for the subsequent contraction of the same `Network` with different tensors.

Using `Network` to perform a tensor contraction includes three steps: 1) Define a network, 2) load the tensors by `PutUniTensor`, and 3) `launch` the contraction. In what follows we briefly introduce how to use `Network`. Further details and a more complicated example of a contraction from an actual tensor network algorithm are discussed in Sec. 8.

In this example, the goal is to use `Network` contraction to multiply the tensors $M_1$ and $M_2$ and sum over the common index $j$, as shown in Fig. 4. One first creates a file `matmul.`

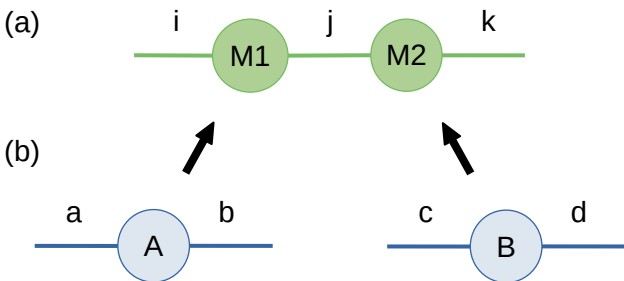

Figure 4: (a) Graphical notation for a matrix-matrix multiplication. (b) Specifying each tensor by an actual tensor object.

net with content as shown in Listing 3. Note that a `Network` is just a blueprint of the tensor contractions. In the code example (Listing 4) below, we first create a `Network` object using the file `matmul.net`. Then, we initialize two rank-2 `UniTensor` A and B. We specify the `UniTensor` objects that are actually to be contracted, by loading them into the `Network` and match the indices by their labels. In this step, the indices have to be assigned in the right order. In particular, we specify A (B) as M1 (M2) in the network. The indices a and b (c and d) are associated to the indices i and j (j and k). After loading the tensors, we launch the contraction and receive the result as a tensor AB whose indices are labeled by i and k.

Note that the indices a, b, c, d of the actual `UniTensor` objects are independent of the indices i, j, k defined in the network. With this design, one can implement an algorithm with meaningful index names, independent of the actual tensors to be contracted later on. On the other hand, a `Network` can be reused and loaded with different tensors and label conventions. As shown in code example Listing 5, the same `Network` object can be used to perform the multiplication BA instead of AB without relabeling.

```
# Network definition saved in file "matmul.net"
M1:  i, j
M2:  j, k
TOUT: i; k
```

Listing 3: Network file "matmul.net" defining a matrix-matrix multiplication.

```
# Create a Network object from Listing 3
net = cytnx.Network("matmul.net")
# Initialize the tensors and their labels
A = cytnx.UniTensor.ones([2,2]).relabels(["a","b"]).set_name("A")
B = cytnx.UniTensor.ones([2,2]).relabels(["c","d"]).set_name("B")
# Put the tensors into the network
net.PutUniTensor("M1", A, ["a", "b"])
net.PutUniTensor("M2", B, ["c", "d"])
# Launch the contraction
AB = net.Launch()
```

Listing 4: Network contraction for matrix-matrix multiplication AB.

```
net.PutUniTensor("M1", B, ["c", "d"])
net.PutUniTensor("M2", A, ["a", "b"])
BA = net.Launch()
```

Listing 5: Network contraction for matrix-matrix multiplication BA.

# 3 Conventions and object behavior

In this section we discuss the naming conventions of Cytnx and the object behavior.

## 3.1 Naming convention

Generally, the function names in Cytnx follow these rules:

- If a function is acting on objects, its name starts with a capital letter. For example, `Contract(A,B)` contracts two tensors `A` and `B`.

- If a function is a member function or a generating function, it starts with a lowercase letter. For example, `UniTensor.permute([1,0])` permutes the indices of a `UniTensor` object, where `permute` is a member function. `uT=cytnx.UniTensor.zeros([2,2])` generates a `UniTensor uT`, so `zeros` is a generating function for the `UniTensor` object.

- Types defined in Cytnx always start with capital letters. For example `bd=cytnx.Bond(10)` creates an object of `Bond` type.

- If a function name does not end with an underscore, it will not change the input object. For example, `B=A.permute([1,0])` creates a new object `B` that is `A` after permutation, while `A` remains unpermuted.

- A function that ends with an underscore indicates one of the following scenarios:

  1. The input object will be changed. For example, `cytnx.linalg.Pow_(A,2)` replaces `A` by $A^2$. Compare with `Asq = cytnx.linalg.Pow(A,2)`, which creates a new tensor $Asq = A^2$ and leaves `A` unchanged.

  2. For a member function, it is an in-place operation. For example, `A.permute_([1,0])` changes `A` by permuting its indices. Member functions which end with an underscore return a reference to themselves.

## 3.2 Object behavior

In order to have a similar syntax and API in C++ and Python and to reduce the redundant memory allocation, all objects in Cytnx are references. The only exceptions are `Accessor` and `LinOp`. In Python, all objects are references by default already. On the C++ side, however, the reference character has to be implemented by Cytnx explicitly. For example, consider a `UniTensor` object in Python and C++ respectively.

```
# Python:
A = cytnx.UniTensor.zeros([2,3])
B = A
print(B is A) # Output: True
```

Listing 6: Python example.

```
// C++
auto A = cytnx::UniTensor::zeros({2,3});
auto B = A;
cout << cytnx::is(B,A) << endl; // Output: true
```

Listing 7: C++ example.

It is apparent that a line to line translation between C++ and Python can be done with minimal effort. To achieve this, we implement the assignment "=" on the C++ side such that *B* is a reference to *A*. This also avoids copying of data. Additionally, Cytnx implements the `is` clause in the C++ API. This way, one can use `is` to check if two objects reference to the same memory. These additional implementations ensure that C++ and Python can have the same behavior.

In case where a copy of an object is needed, one can use `clone` as shown below. The functions `is` and `clone` support all objects of Cytnx.

```python
# Python
A = cytnx.UniTensor.zeros([2,3])
B = A
C = A.clone()
print(B is A) # Output: True
print(C is A) # Output: False
```

Listing 8: Clone an object in Python.

```cpp
// C++
auto A = cytnx::UniTensor::zeros({2,3});
auto B = A;
auto C = A.clone();
cout << cytnx::is(B,A) << endl; // Output: true
cout << cytnx::is(C,A) << endl; // Output: false
```

Listing 9: Clone an object in C++.

Member functions which only change the metadata of a tensor (e.g. `permute`, `reshape`, `relabel`) create a new tensor with independent metadata. However, the new and old tensors share the same memory for the tensor elements. Thus, changing the elements of the new tensor will change the elements of the old one as well and vice versa. Changes to the metadata do not affect the other tensor though. If the tensor elements of the old and new tensors shall be independent of each other, use `clone`. Member functions which end with an underscore (e.g. `permute_`, `reshape_`, `relabel_`) change the tensor directly and return a reference to it, so no independent metadata is created. For example, we permute a tensor and change an element to see the behavior of the functions:

```python
# Python
A = cytnx.UniTensor.zeros([2,3])
B = A.permute([1,0]) # A and B share tensor elements, different metadata
B[0,1] = 1.
C = A.clone().permute([1,0]) # C is independent of A
C[0,1] = 2.
print(A) # Original shape, element A[1,0] == 1.
print(B) # Permuted shape, element B[0,1] == 1.
print(C) # Permuted shape, element C[0,1] == 2.
D = A.permute_([1,0])   # D is a reference to A
print(A) # Permuted shape, element A[0,1] == 1.
print(D is A) # Output: True
```

Listing 10: References and copies of tensor elements and metadata when using member functions.

## 4 Non-symmetric Bonds

In Cytnx, a `Bond` represents an index (or axis, as in the NumPy convention) of a tensor. A `Bond` carries information about the dimension of a given index, which is the number of possible

values the index can take. Moreover, bonds can have a direction, distinguishing incoming and outgoing indices of a tensor. When symmetries shall be preserved, the `Bond` objects carry additional information like quantum numbers associated with the symmetries. Bonds with symmetries are explained in Sec. 6.2. Here, we focus on bonds without symmetry properties. `Bond` objects can be used to initialize a `UniTensor`, see Sec. 5.

For a bond without quantum numbers, one can define a `Bond` object simply by specifying its dimension. Additionally, bonds can be directed and either be incoming or outgoing. If only the bond dimension is specified, the type is always `REGULAR` and thus un-directed. A directed bond can be created with `cytnx.BD_IN` or `cytnx.BD_OUT`. In this case, an outgoing bond can only be contracted with an incoming bond with the same size. When using symmetries, the bonds must be directed.

```
# Create an undirected bond with dimension 10
bond = cytnx.Bond(10)
print(bond)      # Output: Dim = 10 |type: REGULAR
# Create a directed (incoming) bond with dimension 20
bondIn = cytnx.Bond(20, cytnx.BD_IN)
print(bondIn)    # Output: Dim = 20 |type: |IN (KET)>
```

Listing 11: Create an undirected bond or an outgoing bond.

**Combining bonds.** One can combine two bonds to create a new bond. The dimension of the new bond is the product of the dimensions of the input bonds, as illustrated below (Listing 12):

```
bond1 = cytnx.Bond(10)
bond2 = cytnx.Bond(2)
# Combine bond1 and bond2
bond12 = bond1.combineBond(bond2)
print(bond1)     # Output: Dim = 10 |type: REGULAR
print(bond2)     # Output: Dim = 2 |type: REGULAR
print(bond12)    # Output: Dim = 20 |type: REGULAR
```

Listing 12: Combine two bonds to create a new bond.

Moreover, an existing bond can be modified by merging another bond:

```
bond1 = cytnx.Bond(10)
bond2 = cytnx.Bond(2)
# Merge bond2 into bond1
bond1.combineBond_(bond2)
print(bond1)     # Dim = 20 |type: REGULAR
```

Listing 13: Merge one bond into another bond.

It is also possible to combine several bonds to create a new bond, or to modify an existing bond by merging several bonds. In this case, a list of `Bond` objects is used as the argument, as illustrated below (Listing 14):

```
bond1 = cytnx.Bond(2)
bond2 = cytnx.Bond(2)
bond3 = cytnx.Bond(2)
# Merge bond2 and bond 3 with bond1
bond123 = bond1.combineBonds([bond2,bond3])
print(bond123)   # Output: Dim = 8 |type: REGULAR
bond1.combineBonds_([bond2,bond3])
print(bond1)     # Output: Dim = 8 |type: REGULAR
```

Listing 14: Merge multiple bonds.

# 5 Non-Symmteric UniTensor

`UniTensor` is the central object in Cytnx. It is designed to represent all necessary information of a tensor so that tensor manipulations and calculations can be easily done. As indicated in Fig. 5, a `UniTensor` consists of three parts: Block(s), Bond(s), and Label(s). Additionally, each tensor can be given a name. Blocks contain the value of the tensor elements. Internally they are represented by `Tensor` objects (see Appendix A), which are very similar to NumPy.array or PyTorch.tensor. However, in typical situations the user does not need to interact directly with `Tensor` objects. Bonds are represented by `Bond` objects that define dimensions of the indices and can be directed as incoming or outgoing (see Sec. 4).

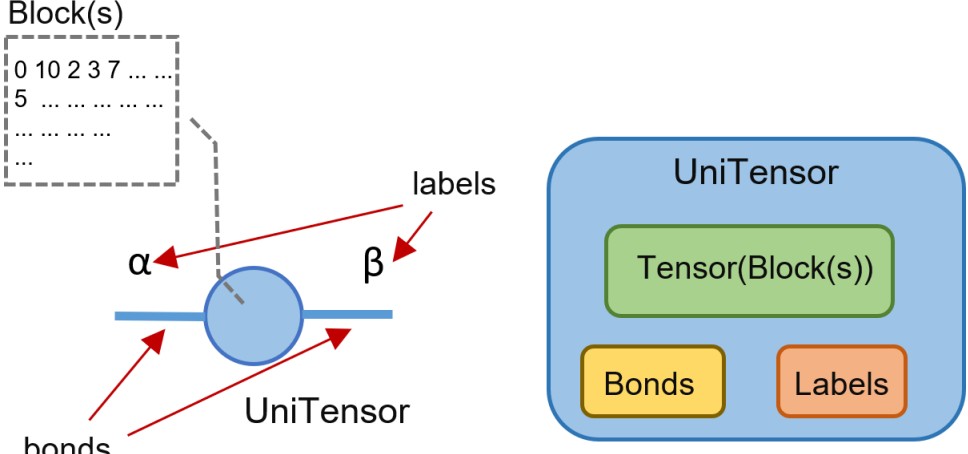

Figure 5: Structure of a `UniTensor`.

## 5.1 Creating a non-symmetric UniTensor

One can create a `UniTensor` through generators such as `zero`, `arange` or `ones` or `eye`. The first argument provides shape information, which is used to construct the `Bond` objects and to determine the rank – the number of tensor indices. Labels can be specified when creating a `UniTensor`, otherwise they are set to be `"0"`, `"1"`, `"2"`, ... by default. Consider a rank-3 tensor as shown in Fig. 6, the corresponding `UniTensor` can be created by the code example below (Listing 15).

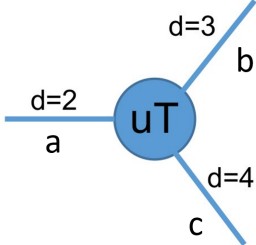

Figure 6: Graphical representation of a tensor. Each line corresponds to one index. The indices have the labels "a", "b" and "c" with dimensions d=2, d=3 and d=4 respectively. "uT": tensor name; Listing 15 shows how to create such a tensor with Cytnx.

```
# Create a rank-3 UniTensor with shape [2,3,4]
uT = cytnx.UniTensor.arange(2*3*4).reshape(2,3,4)\
                                .relabels(["a","b","c"])\
                                .set_name("tensor uT")
```

Listing 15: Create a `UniTensor` by using generators.

Alternatively, we can also create a `UniTensor` by using `Bond` objects as shown in Listing 16. When a `UniTensor` is created from `Bond` objects, it is initialized with zeros by default. A `Bond` within a `UniTensor` can be accessed by `UniTensor.bond`.

```
# Define bonds by their dimensions
bond1 = cytnx.Bond(2)
bond2 = cytnx.Bond(3)
bond3 = cytnx.Bond(4)
# Create a UniTensor by the bonds
uT2 = cytnx.UniTensor([bond1,bond2,bond3])\
                        .relabels(["a","b","c"]).set_name("uT2")
# or, alternative initialization:
uT3 = cytnx.UniTensor([bond1,bond2,bond3], labels=["a","b","c"],\
                        name="uT3")
# Get a bond from a UniTensor
bonda = uT2.bond("a")
```

Listing 16: Create a `UniTensor` from `Bond` objects.

In this example we show two ways to initialize the attributes of the `UniTensor`. We can either use the methods `relabels` to define the labels and `set_name` to set the name of a tensor as for `uT2` in Listing 16. Or we set these properties with the arguments `labels` and `name` in the initialization method as for `uT3` in Listing 16. We recommend the former way where methods are used subsequently. This way, the code can be ported between Python and C++ easily. In some cases (for example in Listing 15), the tensor has to be reshaped as well before the labels can be set, which is straightforward if the properties of the tensor are set by methods instead of initializer arguments.

**Complex elements and data types.** We can create a `UniTensor` with a different data type, for example, with complex elements. This is achieved by setting the `dtype` argument when creating a `UniTensor`:

```
# 1. Create a complex valued UniTensor by initializer
uTc = cytnx.UniTensor.zeros([2,2], dtype=cytnx.Type.ComplexDouble)\
                                .relabels(["a","b"]).set_name("uT complex")
# 2. Create a complex valued UniTensor by Bond
bond1 = cytnx.Bond(2)
bond2 = cytnx.Bond(2)
uTc2 = cytnx.UniTensor([bond1,bond2], dtype=cytnx.Type.ComplexDouble)\
                                .relabels(["bd1","bd2"]).set_name("uT complex 2")
```

Listing 17: Create a `UniTensor` with complex elements.

**Random elements.** A `UniTensor` with random elements can be created easily. The code example below (Listing 18) demonstrates how to initialize a `UniTensor` with elements uniformly in the range [0, 1] or with elements from a Gaussian distribution with mean value 0 and standard deviation 1 respectively.

```
# Create a random tensor with elements uniformly in the range [0, 1]
rT = cytnx.UniTensor.uniform([2,3,2], low=0., high=1.)
```

```
# Create a random tensor with elements from a Gaussian distribution
# with mean value 0 and standard deviation 1
nT = cytnx.UniTensor.normal([2,3,2], mean=0., std=1.)
```

Listing 18: Create `UniTensor` with random elements.

Moreover, one can randomize an existing `UniTensor`. This is particularly useful if a `UniTensor` is created from `Bond` objects, as it is typically the case for symmetric tensors. To obtain a `UniTensor` with random elements, one first creates a `UniTensor`, and then applies `random.uniform_` or `random.normal_` to it:

```
# Supposes uT is a UniTensor that is already created as in Listing 15
# Randomize the elements of uT uniformly in the range [-1, 1]
cytnx.random.uniform_(uT, low=-1., high=1.)
# Randomize the elements of uT according to a normal distribution
# with mean value 2 and standard deviation 3
cytnx.random.normal_(uT, mean=2., std=3.)
```

Listing 19: Randomize the elements of a `UniTensor`.

We note that `random.uniform_` and `random.normal_` work with `UniTensor` with and without symmetry. Real and complex date types of the `UniTensor` are supported. For complex data types, the real and imaginary part are randomized independently of each other.

## 5.2 Viewing/Changing labels

Indices can be addressed by their labels. To view the current labels and their internal order, one can use `UniTensor.labels`. Moreover, it might be necessary to change the labels for some bond(s). To change the label associated with one single bond, one can use `UniTensor.relabel_`, while `UniTensor.relabels_` changes multiple labels associated with a list of bonds. In this code example, we show how to relabel one bond or multiple bonds and verify that the labels are changed correctly:

```
# Create a rank-4 UniTensor with shape [2,3,4,5]
uT = cytnx.UniTensor.ones([2,3,4,5])\
                        .relabels(["a","b","c","d"]).set_name("uT")
print(uT.labels()) # Output: ['a', 'b', 'c', 'd']
# Change the label "a" to "i"
uT.relabel_(old_label="a", new_label="i")
print(uT.labels()) # Output: ['i', 'b', 'c', 'd']
# Change multiple labels
uT.relabels_(["b","c"], ["j","k"])
print(uT.labels()) # Output: ['i', 'j', 'k', 'd']
```

Listing 20: Relabel the bond(s) in a `UniTensor`.

We may want to create a `UniTensor` from an existing one without modifying the original tensor, but we want to use a different set of labels. A typical situation where this is useful is the contraction of tensors by summing over common indices (see Sec. 7). Creating a copy of the tensor data is not desired, since it would double the memory usage. In such cases one can use `relabel(s)` without underscore. This returns a new `UniTensor` object with different metadata (in this case only the labels are changed), but the actual memory block(s) are still referring to the old ones. The arguments of `relabel(s)` are similar to `relabel(s)_`.

```
uT_new = uT.relabel(old_label="d", new_label="l")
```

Users need to pay attention when using `relabel` for creating a `UniTensor`. Since the old and new tensors share the same memory for the data, any change of the elements in the

new `UniTensor` affect the original `UniTensor` as well and vice versa. One can check if two `UniTensor`s share the same data with

```
print(uT_new.same_data(uT)) # Output: True
```

which will print `True` (`False`) if they share (do not share) the same memory for the data.

## 5.3 Permute

When a `UniTensor` is created, two important attributes are defined for each index: its position and label. We consider the code example Listing 15. The three indices `bond1`, `bond2` and `bond3` have positions 0, 1, and 2, and labels `"a"`, `"b"` and `"c"` respectively. The library is designed such that users do not need to worry about the index positions in most applications. Instead, indices can be addressed by their labels. For example, tensor contractions can be defined purely by the index labels. However, the actual data in a `UniTensor` is stored as a multi-dimensional array (a `Tensor` object, see Appendix A) which has a specific order for the index positions. Conceptually, if one changes the index order of a `UniTensor`, it still represents the same tensor, but the way that the data is stored in memory is changed.

To specify the internal order of indices in a `UniTensor`, one can use `UniTensor.permute`. The new order can be defined by the index labels (strings) or the index positions (integers), as shown in the code example below (Listing 21). There, we use `labels` to see the new index order after calling `UniTensor.permute`. Note that the same labels can still be used to find the correct index even after a permutation. Therefore, all methods that use index labels do not need to be changed after a permutation.

```
# Create a UniTensor
uT = cytnx.UniTensor.arange(2*3*4).reshape(2,3,4)\
                        .relabels(["a","b","c"]).set_name("uT")
print(uT.labels())  # Output: ['a', 'b', 'c']
print(uT.shape())   # Output: [2, 3, 4]
# Permute by index labels
uT2 = uT.permute(["b","c","a"])
print(uT2.labels()) # Output: ['b', 'c', 'a']
print(uT2.shape())  # Output: [3, 4, 2]
# Permute by index positions
uT3 = uT.permute([2,0,1])
print(uT3.labels()) # Output: ['c', 'a', 'b']
print(uT3.shape())  # Output: [4, 2, 3]
```

Listing 21: Permute the indices of a `UniTensor`.

## 5.4 Tensor information

Cytnx provides different ways to print information about a `UniTensor`. For the meta information but not the value of the elements, `print_diagram` can be used to visualize a `UniTensor` as a diagram. Consider the following code example and corresponding output (Listing 22):

```
# Supposes uT is a UniTensor that is already created as in Listing 15
# Print the diagram of a UniTensor
uT.print_diagram()

''' -------- Output ---------
tensor Name : tensor uT
tensor Rank : 3
block_form  : False
is_diag     : False
on device   : cytnx device: CPU
```

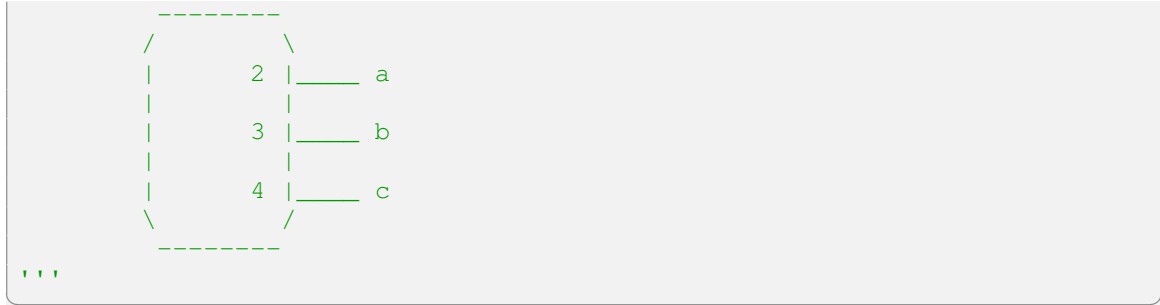

```
          --------
       /            \
      |       2   |____   a
      |           |
      |       3   |____   b
      |           |
      |       4   |____   c
       \          /
         --------
'''
```

Listing 22: Diagram representation of a `UniTensor`.

The command `UniTensor.print_diagram` shows information about the name, rank, block structure, diagonal form, and device.[2] Moreover, it prints a diagram of the tensor. The diagram provides visual information about labels, shapes, and `rowrank`.

The diagram shows that the indices are labeled by `["a", "b", "c"]` with this internal order. They are ordered from upper left to lower left, and then from upper right to lower right. Typically, users do not need to know the internal order of the indices since bonds can be accessed by their labels instead. However, the index order can be used explicitly if needed. The integers inside the tensor show the dimension of the bond, such that one can read the shape `[2, 3, 4]` of the tensor from the diagram.

We observe that bond `"a"` is attached to the left of the tensor, while bonds `"b"` and `"c"` are attached to the right. This is controlled by a property called `rowrank`, which determines how to interpret a tensor as a matrix: The combination of the first `rowrank` bonds are interpreted as the row index, while the combination of the remaining bonds is interpreted as the column index. In the example above, the `rowrank` of the tensor is 1. This means that if we want to interpret this rank-3 tensor as a matrix, we get $uT_{abc} \rightarrow [M]_{(a),(bc)}$, where the matrix $[M]$ has row index $(a)$ and column index $(bc)$.

When a `UniTensor` is created, it acquires certain default `rowrank`. This plays a role *only if* a `UniTensor` is used in a linear algebra function, since it needs to interpret a tensor as a matrix. Consequently, before calling a tensor decomposition function, one may need to permute the labels and set the proper `rowrank` to ensure the correction mapping to a matrix. More details will be discussed in Sec. 9.

Moreover, one can use `print` to see the tensor elements as an $N$-dimensional array:

```
# Supposes uT is a UniTensor that is already created as in Listing 15
print(uT)

''' -------- Output ---------
Tensor name: tensor uT
is_diag    : False
contiguous : True

Total elem: 24
type  : Double (Float64)
cytnx device: CPU
Shape : (2,3,4)
[[[0.00000e+00 1.00000e+00 2.00000e+00 3.00000e+00 ]
  [4.00000e+00 5.00000e+00 6.00000e+00 7.00000e+00 ]
  [8.00000e+00 9.00000e+00 1.00000e+01 1.10000e+01 ]]
 [[1.20000e+01 1.30000e+01 1.40000e+01 1.50000e+01 ]
  [1.60000e+01 1.70000e+01 1.80000e+01 1.90000e+01 ]
  [2.00000e+01 2.10000e+01 2.20000e+01 2.30000e+01 ]]]
```

---

[2]name: can be set with `set_name`; rank: number of indices; block structure: used for a symmetric `UniTensor`, see Sec. 6; diagonal form: used to efficiently store diagonal matrices like the identity, singular values or eigenvalues; device: the data can be stored on the CPU or GPU memory, see Sec. 10 for details.

```
'''
```

Listing 23: Print a `UniTensor`.

## 5.5 Arithmetic operations and tensor operations

Cytnx supports basic addition (+), subtraction (−), multiplication (∗), and division (/) operations for `UniTensor`s. Note that all of these operators are *element-wise*, which means that the operation is applied between each pair of elements from the two `UniTensor`s. Obviously, the two operated `UniTensor`s must have the same shape. Other useful operations are `UniTensor.Norm`, which returns the two-norm of the tensor and `UniTensor.Conj`, which takes the complex conjugate of all elements (see Sec. 6.4 for an example). An extensive list of tensor operations and linear algebra functions is provided in the User Guide [39]. Further linear algebra functions are discussed in Sec. 9.

## 5.6 Getting/Setting elements

**`UniTensor.at()`.** To get or set a single element in a `UniTensor`, one can call `UniTensor.at`. It returns a *proxy* which contains a *reference* to the element, and therefore can be used to get or set the element. The first argument of `UniTensor.at` contains the labels of the bonds in the `UniTensor`, and the second argument contains the corresponding index values for the element one wants to access. For example:

```
# Create a UniTensor
uT = cytnx.UniTensor.zeros([3,3]).relabels(["a","b"]).set_name("uT")
# Print the element of the UniTensor
print(uT.at(["a","b"],[0,2]).value) # Output: 0.0
# Set the element to a new value and see the change
uT.at(["a","b"], [0,2]).value = -1
print(uT.at(["a","b"],[0,2]).value) # Output: -1.0
```

Listing 24: Get or set a single element.

**Slicing.** To get or set a range of elements in a `UniTensor` one can use slicing, where the internal order of indices is used.

```
# Create a UniTensor
uT=cytnx.UniTensor.zeros([2,3,4]).set_name("uT")\
                        .relabels(["a", "b", "c"])
# set and get an element
uT[1,1,2] = 3.0
print(uT[1,1,2])     # Output: 3.0
# get a slice of a UniTensor
uTpart = uT[0,:,1:3]
# set a slice of a UniTensor
uT[0,:,1:3]=cytnx.UniTensor.ones([3,2])
# print the original slice
print(uTpart)
# print the UniTensor
print(uT)

''' -------- Output ---------
-------- start of print ---------
Tensor name:
is_diag    : False
contiguous : True
```

```
Total elem: 1
type  : Double (Float64)
cytnx device: CPU
Shape : (1)
[3.00000e+00 ]

-------- start of print ---------
Tensor name:
is_diag    : False
contiguous : True

Total elem: 6
type  : Double (Float64)
cytnx device: CPU
Shape : (3,2)
[[0.00000e+00 0.00000e+00 ]
 [0.00000e+00 0.00000e+00 ]
 [0.00000e+00 0.00000e+00 ]]

-------- start of print ---------
Tensor name: uT
is_diag    : False
contiguous : True

Total elem: 24
type  : Double (Float64)
cytnx device: CPU
Shape : (2,3,4)
[[[0.00000e+00 1.00000e+00 1.00000e+00 0.00000e+00 ]
   [0.00000e+00 1.00000e+00 1.00000e+00 0.00000e+00 ]
   [0.00000e+00 1.00000e+00 1.00000e+00 0.00000e+00 ]]
 [[0.00000e+00 0.00000e+00 0.00000e+00 0.00000e+00 ]
   [0.00000e+00 0.00000e+00 3.00000e+00 0.00000e+00 ]
   [0.00000e+00 0.00000e+00 0.00000e+00 0.00000e+00 ]]]
'''
```

Listing 25: Get or set a slice.

## 5.7 Accessing data in a block

The elements of a `UniTensor` are stored in a `Tensor` object. One can access and manipulate the `Tensor` as described in Appendix A and read and change the data of the `UniTensor` this way.

**Getting a block.**   The `Tensor` can be accessed by `UniTensor.get_block` and `UniTensor.get_block_`, where the first method returns a copy of the `Tensor` object in the `UniTensor`, while the second method returns a reference. As an example, we manipulate the `Tensor` in order to change an element of the `UniTensor`:

```
uT = cytnx.UniTensor.ones([3,3]).set_name("uT")
B = uT.get_block_()
B[0,0] = 0.
print(uT)

''' -------- Output ---------
Tensor name: uT
is_diag    : False
contiguous : True
```

```
Total elem: 9
type  : Double (Float64)
cytnx device: CPU
Shape : (3,3)
[[0.00000e+00 1.00000e+00 1.00000e+00 ]
 [1.00000e+00 1.00000e+00 1.00000e+00 ]
 [1.00000e+00 1.00000e+00 1.00000e+00 ]]
'''
```

Listing 26: Get a block (`Tensor`) from a non-symmetric `UniTensor`.

A `UniTensor` without symmetries contains only one block, while several blocks exist when symmetries are involved, see Sec. 6.3.3.

**Putting a block.** The methods `UniTensor.put_block()` and `UniTensor.put_block_()` replace the data of a `UniTensor` by a `Tensor`. For example, after getting a block from a `UniTensor`, we can manipulate the corresponding `Tensor` object and write it back to the `UniTensor`:

```
uT = cytnx.UniTensor.ones([3,3]).set_name("uT")
B = uT.get_block()
# Manipulate the Tensor
B[0,0] = 0.
# Put the Tensor back to the UniTensor
uT.put_block(B)
print(uT)
```

Listing 27: Replace a block of a `UniTensor` by a `Tensor`.

We get the same output as in Listing 26, where we manipulated the reference to the block directly. Note that the shape of the `Tensor` has to match that of the `UniTensor` whose data shall be replaced.

## 5.8 Converting to/from a NumPy array

It is very easy to create a `UniTensor` from a NumPy array and vice versa. This is particularly useful for including existing code or when one wants to quickly test an algorithm using Python and then convert to Cytnx for acceleration, quantum numbers, or larger-scale applications. Such flexibility also allows users to make use of the tools from other packages like NumPy, SciPy and PyTorch. Since all popular tensor libraries allow for conversion of their containers to a NumPy array, the latter serves as a bridge to Cytnx. In the future, we plan to support PyTorch directly as the underlying container of a tensor in Cytnx. This way, users will be able to use automatic differentiation directly in Cytnx.

In Listing 28 we show an example where we create a NumPy array and then convert it to a `UniTensor`. The tensor corresponds to the matrix product operator that defines the Hamiltonian of the transverse-field Ising model [40].

```
import numpy as np
# Define operators
Sx = np.array([[0.,1.],[1.,0.]])
Sz = np.array([[1.,0.],[0.,-1.]])
I  = np.array([[1.,0.],[0.,1.]])
# Define MPO tensor
hx, hz = 0.5, 0.5       # longitudinal and transverse fields
M = np.zeros((3,3,2,2)) # The first two indices are virtual indices,
                        # and the last two are physical indices
M[0,0] = I
M[2,2] = I
```

```
M[1,0] = Sx
M[2,1] = -Sx
M[2,0] = -hz*Sz - hx*Sx
M = cytnx.from_numpy(M) # NumPy array to a Tensor (see Appendix A)
M = cytnx.UniTensor (M) # Tensor to a UniTensor
M.set_name("M").relabels_(["mpoL", "mpoR", "physKet", "physBra"])
M.print_diagram()        # Check diagram
# Create a NumPy array from a non-symmetric UniTensor
T = M.get_block().numpy()
```

Listing 28: Create the tensor of a matrix-product-operator for the transverse-field Ising model as a NumPy array and then convert it to a UniTensor.

# 6 UniTensor with Symmetries

In physics, systems are often symmetric under certain transformations. Exploiting such symmetries can be advantageous in many cases. Cytnx allows for the incorporation of the symmetries on the level of the tensors directly. We explain the basic concepts that are used in Cytnx. Further details on symmetries in tensor network algorithms are found in [41, 42].

According to Noether's theorem, symmetries imply conserved quantities, for example, the total charge in the system. In that case, the Hamiltonian will have a block-diagonal structure where each symmetry sector has a definite charge (quantum number), which is caused by the fact that different charge sectors do not mix with each other. When the symmetric structure is imposed on the level of the tensors which describe the system, they become block-diagonal as well. This substantially reduces the memory costs and the number of variational parameters in numerical algorithms. Thus, larger system sizes or larger bond dimensions are accessible with the same computational costs. Moreover, numerical errors contributed by the elements that do not conserve symmetry (and hence are zero) can be avoided. Finally, simulations can be restricted to certain symmetry sectors if needed.

A quantum-number conserving tensor can be understood in the following way. By definition, each element in a tensor can be labeled by its index value for each index. For example `T[0,0,0]` is an element with index values 0 for all three indices. Consider that in addition to the integer, each index value also carries a quantum number. This is shown in Fig. 7, where bonds are indicated by directed arrows, and each index value, represented by a horizontal line, has an associated quantum number.

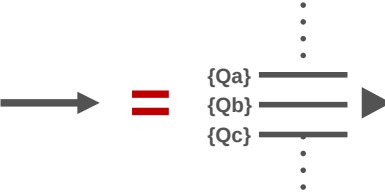

Figure 7: Sketch of a bond with quantum numbers.

One can then define a *quantum-number flux* for each element by adding all the incoming quantum numbers and subtracting all the outgoing quantum numbers. Note that the rule of "addition" and "subtraction" actually depends on the symmetry group associated with the quantum number. For charge conservation, which is associated to a $U(1)$ symmetry, addition and subtraction follow the same rule as for integers. The conservation of a quantum number is ensured by restricting the tensors to only have non-zero elements with *zero flux*. For these elements, the total quantum number flowing into the tensor has to be equal to the total quantum

number flowing out. All other elements that do not conserve the charge must be zero. For a `UniTensor`, these zero-elements are not stored in memory. Therefore, a symmetric tensor can be thought of as a direct sum of blocks of different quantum-number sectors, as represented in Fig. 8.

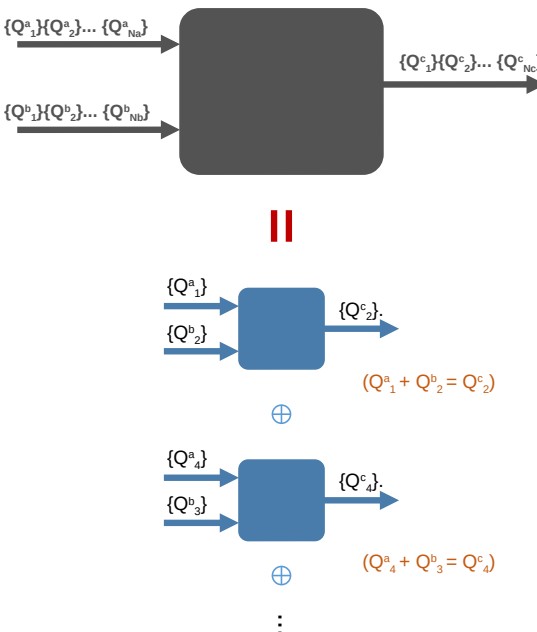

Figure 8: Block structure for a tensor with quantum numbers.

In Cytnx, symmetric `UniTensor`s contain `Bond` objects which carry quantum numbers. The bonds themselves contain a `Symmetry` object describing all the symmetries involved. Currently, abelian symmetries are implemented in Cytnx. Non-abelian symmetries like $SU(2)$ are planed to be supported in the future [43]. Cytnx has predefined symmetry objects for $U(1)$ and $Z_n$ symmetries. Once a symmetric `UniTensor` is defined, it can be used just like before, with typically no changes in the API of functions. In particular, the linear algebra operations respect the block structure automatically.

Sometimes a quantum number flux which differs from zero can be desired. For example, an external field can change the flux locally. In these cases, an additional bond can be attached to the `UniTensor`, which has a dimension of one and carries the quantum number of the flux change. The code in Listing 45 shows an example where this is used for particle number changing operators.

## 6.1 Symmetry and quantum numbers

The symmetry properties are contained in the `Symmetry` class and `Symmetry` objects are used in the construction of a `Bond` with quantum numbers. When a `Symmetry` object is printed, the name and type of the symmetry are displayed, as well as the combine rule and the reverse rule. The combine rule defines how two quantum numbers are merged to a new quantum number. For the additive quantum numbers of $U(1)$, the combine rule is simply the sum of the individual quantum numbers. The reverse rule defines how a quantum number is transformed when the bond is redirected. Quantum numbers on outgoing bonds have the reverse rule applied when they are combined with other quantum numbers. With this procedure, the symmetry preserving entries of a tensor are those where the combination of all quantum numbers is zero. For example, the inverse rule of $U(1)$ takes the negative value of the original quantum number.

For example, we can create a `Symmetry` describing the group $U(1)$ or $Z_3$:

```
U1 = cytnx.Symmetry.U1()
print(U1)

''' -------- Output ---------
[Symmetry]
type : Abelian, U1
combine rule : Q1 + Q2
reverse rule : Q*(-1)
'''
```

Listing 29: Create a $U(1)$ `Symmetry` object.

```
Z3 = cytnx.Symmetry.Zn(3)
print(Z3)

''' -------- Output ---------
[Symmetry]
type : Abelian, Z(3)
combine rule : (Q1 + Q2)%3
reverse rule : Q*(-1)
'''
```

Listing 30: Create a $Z_3$ `Symmetry` object.

## 6.2 Bond with quantum numbers

In Cytnx, the quantum number information is stored in the `Bond` objects. A `Bond` with quantum numbers must have the following properties defined:

1. A direction. The bond is either incoming (`BD_IN`) or outgoing (`BD_OUT`).

2. A list of quantum numbers and their dimensions. Quantum numbers are integers.

3. The corresponding symmetry group(s), see Sec. 6.1.

A direction is essential for any `Bond` in a symmetric `UniTensor`. Symmetry preserving tensors have a block structure, which is determined by the requirement that for non-zero elements the total quantum numbers of the incoming bonds equals the total quantum number of the outgoing bonds. Thus, all entries that do not fulfill this condition have to vanish and the corresponding blocks do not need to be stored explicitly. By defining bonds with quantum numbers, a `UniTensor` created with these bonds will automatically have the desired block structure.

### 6.2.1 Creating bonds with quantum numbers

We consider as an example a tensor with $U(1)$ symmetry. We would like to construct a bond for which the subspace with quantum number $q = 2$ has a dimension (or degeneracy) of 3, while the $q = 4$ subspace has dimension 5. We can create the bond using a list to describe the quantum numbers `[[2],[4]]` and another list containing the dimensions `[3,5]`. Note that the quantum number is enclosed in a list to accommodate multiple symmetries. Instead of the two lists, we can also use a single list of tuples where each tuple combines the information of quantum number and dimension `[([2],3),([4],5)]`. Cytnx also includes a helper function `Qs` to allow for a simpler and well readable syntax. The code `[cytnx.Qs(2)>>3, cytnx.Qs(4)>>5]` generates the aforementioned list of tuples.

```
#Creating a Bond from a list of quantum numbers and a list of dimensions
bond = cytnx.Bond(cytnx.BD_IN, [[2],[4]], [3,5], [cytnx.Symmetry.U1()])
#Creating the same Bond from a tuple of quantum numbers and dimensions
bond = cytnx.Bond(cytnx.BD_IN, [([2],3),([4],5)], [cytnx.Symmetry.U1()])
#Creating the same Bond with the helper function Qs
bond = cytnx.Bond(cytnx.BD_IN,\
                  [cytnx.Qs(2)>>3,cytnx.Qs(4)>>5],[cytnx.Symmetry.U1()])
```

Listing 31: Create a bond with a $U(1)$ quantum number.

The symmetries and the corresponding quantum numbers are encoded by lists. This way, multiple symmetries can be present at the same time. Consider a tensor with $U(1) \otimes Z_2$ symmetry. We specify the symmetry by `[Symmetry.U1(), Symmetry.Zn(2)]` and the quantum numbers by `[q_U1, q_Z2]`. We can use `Qs(q_U1, q_Z2)>>dim` to construct the tuple containing the quantum numbers and the dimension of the subspace. One can print a `bond` to check its quantum numbers.

```
# This creates an incoming Bond
# with U(1)xZ2 quantum numbers qn_U1 and qn_Z2.
# The sector with qn_U1=2 and qn_Z2=0 has dimension 3.
# The sector with qn_U1=4 and qn_Z2=1 has dimension 5.
bond = cytnx.Bond(cytnx.BD_IN,\
                  [cytnx.Qs(2,0)>>3, cytnx.Qs(4,1)>>5],\
                  [cytnx.Symmetry.U1(), cytnx.Symmetry.Zn(2)])
print(bond)

''' -------- Output ---------
Dim = 8 |type: |IN (KET)>
 U1::   +2  +4
 Z2::   +0  +1
Deg>>    3   5
'''
```

Listing 32: Create a bond with two quantum numbers.

### 6.2.2 Combining bonds with quantum numbers

Bonds with symmetries can be combined. The new quantum numbers are the combination of the two input quantum numbers. The actual combine rule depends on the corresponding symmetry group. For $U(1)$ for example, two quantum numbers are combined by adding them. In general, different pairs of quantum numbers can be combined to the same quantum number. For example, both $0 + 3$ and $1 + 2$ give the quantum number 3 in the $U(1)$ case. Therefore, the output bond in general can have several quantum-number sectors with the same quantum number. By default, the sectors with the same quantum numbers will be grouped into a single sector with accordingly increased dimension. However, users can choose not to group them by setting the argument `is_grp=False`. Here is an example of how to combine two bonds with `is_grp=True` (default) and `is_grp=False`:

```
bond1 = cytnx.Bond(cytnx.BD_IN,\
                   [cytnx.Qs(0)>>1,cytnx.Qs(1)>>1], [cytnx.Symmetry.U1()])
bond2 = cytnx.Bond(cytnx.BD_IN,\
                   [cytnx.Qs(2)>>1,cytnx.Qs(3)>>1], [cytnx.Symmetry.U1()])
bond12 = bond1.combineBond(bond2)
bond12_no_group = bond1.combineBond(bond2,is_grp=False)
print('bond1:\n',bond1)
print('bond2:\n',bond2)
print('bond12:\n',bond12)
print('bond12_no_group:\n',bond12_no_group)
```

```
''' -------- Output ---------
bond1:
 Dim = 2 |type: |IN (KET)>
 U1::   +0  +1
Deg>>    1   1

bond2:
 Dim = 2 |type: |IN (KET)>
 U1::   +2  +3
Deg>>    1   1

bond12:
 Dim = 4 |type: |IN (KET)>
 U1::   +2  +3  +4
Deg>>    1   2   1

bond12_no_group:
 Dim = 4 |type: |IN (KET)>
 U1::   +2  +3  +3  +4
Deg>>    1   1   1   1
'''
```

Listing 33: Combine bonds with quantum numbers.

In the example we print the `Bond` objects to check their quantum numbers and the corresponding dimensions. We can see that the output quantum numbers are the sums of the input ones, and the sectors with the same quantum numbers are grouped or not depending on the argument `is_grp=False`. Note that the number of sectors (3 for `bond12` and 4 for `bond12_no_group`) can differ if `is_grp` is changed.

Note that only `Bond`s with the same direction can be combined, and the resulting `Bond` will have the same direction. In a physical interpretation, an incoming or outgoing `Bond` can be used to define the Hilbert space for a ket or a bra state, respectively. In this sense, combining two `Bond`s represents a product of Hilbert spaces for the two states, which is well defined only if both states are ket or bra states.

### 6.2.3 Redirecting bonds

One can change the bond direction by using `Bond.redirect` or `Bond.redirect_`:

```
# Supposes bond is a Bond that is already created as in Listing 32
print(bond)      # Output: Dim = 8 |type: |IN (KET)>
# Create a new bond that has the direction changed.
bond2 = bond.redirect()
# Change the direction of a bond
bond.redirect_()
print(bond)      # Output: Dim = 8 |type: < OUT (BRA)|
print(bond2)     # Output: Dim = 8 |type: < OUT (BRA)|
```

Listing 34: Redirect a `Bond`.

## 6.3 Symmetric UniTensor

### 6.3.1 Creating a symmetric UniTensor

A symmetric `UniTensor` is created with `Bond` objects which carry quantum numbers. Conceptually, one needs to identify the symmetries (for example, $U(1)$ symmetry) in the system, and then specify the dimensions of all quantum number sectors for the bonds. As an example,

we consider a rank-3 tensor with $U(1)$ symmetry. The first two bonds are incoming bonds with two quantum number sectors 1 and $-1$, both with dimension 1. The third bond is an outgoing bond with three quantum number sectors 2, 0, and $-2$, with dimensions 1, 2, and 1 respectively. We then initialize a `UniTensor` using these three bonds.

```python
# Define the bonds with quantum numbers
bond1 = cytnx.Bond(cytnx.BD_IN,\
                [cytnx.Qs(1)>>1, cytnx.Qs(-1)>>1],[cytnx.Symmetry.U1()])
bond2 = cytnx.Bond(cytnx.BD_IN,\
                [cytnx.Qs(1)>>1, cytnx.Qs(-1)>>1],[cytnx.Symmetry.U1()])
bond3 = cytnx.Bond(cytnx.BD_OUT,\
                [cytnx.Qs(2)>>1, cytnx.Qs(0)>>2, cytnx.Qs(-2)>>1],\
                [cytnx.Symmetry.U1()])
# Create a symmetric UniTensor with these bonds
uTsym = cytnx.UniTensor([bond1, bond2, bond3]).relabels(["a","b","c"])\
                        .set_name("uTsym")
```

Listing 35: Create a symmetric `UniTensor` using bonds with quantum numbers.

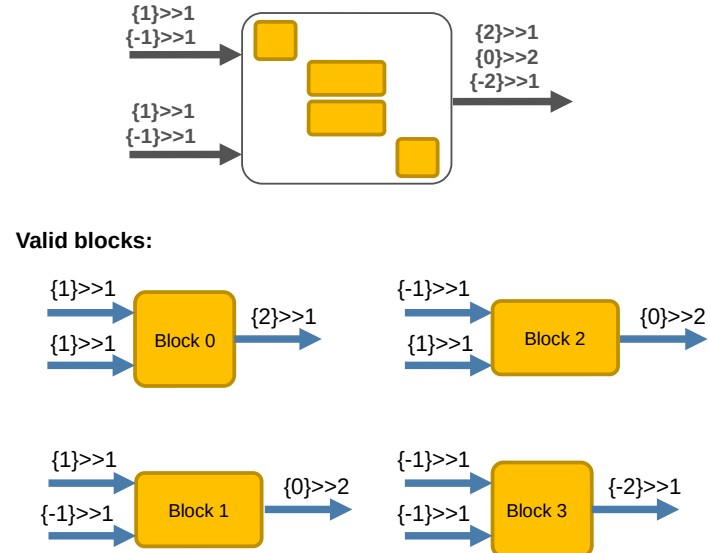

Figure 9: Schematic representation of the block structure of a symmetric tensor. See Listing 35 for the creation of this tensor in Cytnx.

For this `UniTensor`, there are 4 valid blocks that carry zero-flux and thus can have non-zero elements that do not break the symmetry. Figure 9 shows the block structure and the corresponding quantum numbers for each block. We can use `UniTensor.print_blocks` to see all the blocks and their corresponding quantum numbers:

```python
# Supposes uTsym is a UniTensor that is already created as in Listing 35
uTsym.print_blocks()

''' -------- Output ---------
-------- start of print ---------
Tensor name: uTsym
braket_form : True
is_diag     : False
[OVERALL] contiguous : True
========================
BLOCK [#0]
 |- []   : Qn index
 |- Sym(): Qnum of correspond symmetry
```

```
                      ----------
                      |        |
     [0] U1(1)   -->| 1      1 |-->   [0] U1(2)
                      |        |
     [0] U1(1)   -->| 1        |
                      |        |
                      ----------

Total elem: 1
type  : Double (Float64)
cytnx device: CPU
Shape : (1,1,1)
[[[0.00000e+00 ]]]

========================
BLOCK [#1]
 |- []    : Qn index
 |- Sym(): Qnum of correspond symmetry
                      ----------
                      |        |
     [0] U1(1)   -->| 1      2 |-->   [1] U1(0)
                      |        |
     [1] U1(-1)  -->| 1        |
                      |        |
                      ----------

Total elem: 2
type  : Double (Float64)
cytnx device: CPU
Shape : (1,1,2)
[[[0.00000e+00 0.00000e+00 ]]]

========================
BLOCK [#2]
 |- []    : Qn index
 |- Sym(): Qnum of correspond symmetry
                      ----------
                      |        |
     [1] U1(-1)  -->| 1      2 |-->   [1] U1(0)
                      |        |
     [0] U1(1)   -->| 1        |
                      |        |
                      ----------

Total elem: 2
type  : Double (Float64)
cytnx device: CPU
Shape : (1,1,2)
[[[0.00000e+00 0.00000e+00 ]]]

========================
BLOCK [#3]
 |- []    : Qn index
 |- Sym(): Qnum of correspond symmetry
                      ----------
                      |        |
     [1] U1(-1)  -->| 1      1 |-->   [2] U1(-2)
                      |        |
     [1] U1(-1)  -->| 1        |
                      |        |
                      ----------
```

```
Total elem: 1
type  : Double (Float64)
cytnx device: CPU
Shape : (1,1,1)
[[[0.00000e+00 ]]]
'''
```

Listing 36: Print the blocks of a symmetric `UniTensor`.

The *block index* is printed as `BLOCK [#]`, and the number in the square bracket `[]` beside each bond is called a *Qn index* (quantum number index). Qn indices are used to access the sectors in a bond. Consider the third bond (the out-going bond) as an example. It has three possible values for the quantum number: 2, 0, and −2. It is created with this particular ordering of the quantum numbers. Hence, Qn=0 corresponds to the quantum number 2, Qn=1 to the quantum number 0, and Qn=2 to the quantum number −2. Both the block index and the Qn indices can be used to access a block in a `UniTensor` as it will be described below in Sec. 6.3.3.

### 6.3.2   Accessing elements of a symmetric tensor

The `UniTensor.at` method works in the same way as without symmetries (see Sec. 5.6). For a symmetric `UniTensor`, some elements may not exist due to the quantum number constraint. If one tries to access an element that does not correspond to a valid block, an error is thrown:

```
# Supposes uTsym is a UniTensor that is already created as in Listing 35
# Print an invalid element. This will raise an error.
print(uTsym.at([0,0,1]).value)

''' -------- Output ---------
ValueError: [ERROR] trying access an element that is not exists!, using
    T.if_exists = sth or checking with T.exists() to verify before access
     element!
'''
```

Listing 37: Example of accessing a non-existing element in a `UniTensor`. An error is thrown in this case.

However, one can use the proxy to check whether an element corresponds to a valid block with symmetries:

```
# Supposes uTsym is a UniTensor that is already created as in Listing 35
print(uTsym.at([0,0,0]).exists()) # Output: True
print(uTsym.at([0,0,1]).exists()) # Output: False
```

Listing 38: Check if an element is valid in a symmetric `UniTensor`.

This way, one can make sure that an element is part of a valid block before the data is accessed.

### 6.3.3   Accessing blocks of a symmetric tensor

**Getting a block.**   A symmetric `UniTensor` contains multiple blocks for different quantum number sectors in general. We can use the methods `get_block` and `get_block_` to access the blocks. Compared to the non-symmetric case in Sec. 5.7, her we have to specify the block for a symmetric `UniTensor`. Note that `get_block` returns a copy of the `Tensor` object, while `get_block_` returns a reference. We use the $U(1)$ symmetric `UniTensor` from the example in Listing 35 to demonstrate how to access blocks. There are two ways to get a certain block from a symmetric `UniTensor`: by its Qn indices or by its block index. When using Qn indices, the labels of the indices can be given as the first argument of `get_block`, followed by a list of the Qn indices associated with the block.

```
# Supposes uTsym is a UniTensor that is already created as in Listing 35
# Get a block by its Qn indices
# The first argument specifies the labels of the indices,
# and the second argument the corresponding Qn indices
B1 = uTsym.get_block_(["a","b","c"],[0,1,1])
# Get a block by its block index
B2 = uTsym.get_block_(1)
```

Listing 39: Access a block of a symmetric `UniTensor`.

One can also access all valid blocks in a symmetric `UniTensor` with `get_blocks` or `get_blocks_`. These methods return a list (or a vector in C++) of blocks, where each block is a `Tensor` object. The blocks are ordered with ascending block index, corresponding to the order in which they are stored in the `UniTensor`. This order can be checked by `UniTensor.print_blocks()`.

```
# Supposes uTsym is a UniTensor that is already created as in Listing 35
# Get all the blocks in a UniTensor
Blks = uTsym.get_blocks_()
# Print the number of blocks
print(len(Blks))    # Output: 4
# Print each block
print( *Blks )

''' -------- Output ---------

Total elem: 1
type  : Double (Float64)
cytnx device: CPU
Shape : (1,1,1)
[[[0.00000e+00 ]]]

Total elem: 2
type  : Double (Float64)
cytnx device: CPU
Shape : (1,1,2)
[[[0.00000e+00 0.00000e+00 ]]]

Total elem: 2
type  : Double (Float64)
cytnx device: CPU
Shape : (1,1,2)
[[[0.00000e+00 0.00000e+00 ]]]

Total elem: 1
type  : Double (Float64)
cytnx device: CPU
Shape : (1,1,1)
[[[0.00000e+00 ]]]
'''
```

Listing 40: Get all the blocks in a `UniTensor`.

**Putting a block.**  Similarly to the non-symmetric case, we can put a `Tensor` as a block to a `UniTensor`. In the symmetric case, one can put a `Tensor` to a particular block by specifying the Qn indices or the block index:

```
# Supposes uTsym is a UniTensor that is already created as in Listing 35
# Get a block as a Tensor object by Qn indices
```

```
B1 = uTsym.get_block([0,1,1])
# Manipulate the Tensor
B1[0,0,0] = 1.
# Put the Tensor to a block by Qn indices
uTsym.put_block(B1,[0,1,1])
# Put the Tensor to a block by block index
uTsym.put_block(B1,1)
print(uTsym.get_block(1))

''' -------- Output ---------
Total elem: 2
type  : Double (Float64)
cytnx device: CPU
Shape : (1,1,2)
[[[1.00000e+00 0.00000e+00 ]]]
'''
```

Listing 41: Put a block into a `UniTensor`.

## 6.4 Change the bond direction

One can change the direction of all bonds with `UniTensor.Transpose` or `UniTensor.Dagger`. The latter returns the Hermitian conjugate of a `UniTensor`, with all bond directions inverted and the complex conjugate of all elements taken. For tensors with non-directional bonds (`REGULAR` type), the bonds remain non-directional after the operation. If only the complex conjugate of a tensor is desired, `UniTensor.Conj` can be used. If the `dtype` of the tensor corresponds to real valued elements, the complex conjugation has no effect. In line with the general naming convention, `UniTensor.Transpose_`, `UniTensor.Conj_` and `UniTensor.Dagger_` change the tensor in-place. The following code shows an example for the `UniTensor.Transpose`, `UniTensor.Conj` and `UniTensor.Dagger` operations:

```
# Create bonds
bond1 = cytnx.Bond(1,cytnx.BD_IN)
bond2 = bond1.redirect()
# Create a symmetric UniTensor with these bonds
uTc = cytnx.UniTensor([bond1,bond2], dtype=cytnx.Type.ComplexDouble)\
                      .relabels(["a", "b"]).set_name("uTc")
# Set an element
uTc.at([0,0]).value = 1.+2.j
# Change the direction of the bonds
uTtransp = uTc.Transpose().set_name("uTc transpose")
# Conjugate the elements
uTconj = uTc.Conj().set_name("uTc conjugate")
# Take the Hermitian conjugate
uTdag = uTc.Dagger().set_name("uTc Hermitian conjugate")
# Equivalently:
uTdag2 = uTc.Conj().Transpose().set_name("uTc conjugate transpose")
print(uTc)      # Part of output: [[1.00000e+00+2.00000e+00j ]]
print(uTconj)   # Part of output: [[1.00000e+00-2.00000e+00j ]]
print(uTtransp) # Part of output: [[1.00000e+00+2.00000e+00j ]]
print(uTdag)    # Part of output: [[1.00000e+00-2.00000e+00j ]]
```

Listing 42: Transpose, conjugate and Hermitian conjugate of a tensor.

## 6.5 Conversion between UniTensor with and without symmetries

One can use `UniTensor.convert_from` to perform conversion between `UniTensor`s with and without symmetries. If one tries to convert a non-symmetric `UniTensor` with non-zero

elements in the symmetry violating blocks to a symmetric one, an error is thrown. It is possible to overwrite the check by using keyword argument `force=True`. In this case, the elements in the symmetry violating blocks are ignored.

As an example, we first create a non-symmetric `UniTensor`, which represents the two-site Hamiltonian $H = \sigma_1^+ \sigma_2^- + \sigma_1^- \sigma_2^+ + \sigma_1^z \sigma_2^z$. We initialize the $\sigma$-matrices, take Kronecker products of pairs of them, and then sum the terms to get the Hamiltonian:

```python
# Create sigma matrices as non-symmetric UniTensors
Sp = cytnx.UniTensor.zeros([2,2]) # sigma^+
Sp[0, 1] = +1.
Sm = cytnx.UniTensor.zeros([2,2]) # sigma^-
Sm[1, 0] = +1.
Sz = cytnx.UniTensor.zeros([2,2]) # sigma^z
Sz[0, 0] = +1.
Sz[1, 1] = -1.
# Create two copies of all sigma matrices for the two sites
Sp1 = Sp.clone().set_name("Sp1").relabels(["i1","j1"])
Sp2 = Sp.clone().set_name("Sp2").relabels(["i2","j2"])
Sm1 = Sm.clone().set_name("Sm1").relabels(["i1","j1"])
Sm2 = Sm.clone().set_name("Sm2").relabels(["i2","j2"])
Sz1 = Sz.clone().set_name("Sz1").relabels(["i1","j1"])
Sz2 = Sz.clone().set_name("Sz2").relabels(["i2","j2"])
# Kronecker product of two sigma matrices at different sites
Sp1Sm2 = cytnx.Contract(Sp1,Sm2).set_name("Sp1Sm2")
Sp1Sm2.permute_(["i1","i2","j1","j2"])
Sm1Sp2 = cytnx.Contract(Sm1,Sp2).set_name("Sm1Sp2")
Sm1Sp2.permute_(["i1","i2","j1","j2"])
Sz1Sz2 = cytnx.Contract(Sz1,Sz2).set_name("Sz1Sz2")
Sz1Sz2.permute_(["i1","i2","j1","j2"])
# Create Hamiltonian as non-symmetric UniTensor
H = Sp1Sm2 + Sm1Sp2 + Sz1Sz2
H.set_name("H non-Symmetric").relabels_(["i1","i2","j1","j2"])
# Print in matrix form to check
print(H.reshape(4,4))
```

Listing 43: Create a two-site Hamiltonian $H = \sigma_1^+ \sigma_2^- + \sigma_1^- \sigma_2^+ + \sigma_1^z \sigma_2^z$ as a non-symmetric `UniTensor`.

This non-symmetric `UniTensor` can be converted to a symmetric one, which has to be initialized with the correct quantum numbers. In our case, we create a $U(1)$ symmetric `UniTensor` and copy data from the non-symmetric UniTensor:

```python
# Supposes H is a UniTensor that is already created as in Listing 43
# Create bonds with U(1) symmetry
bi = cytnx.Bond(cytnx.BD_IN, [cytnx.Qs(+1)>>1, cytnx.Qs(-1)>>1],\
                [cytnx.Symmetry.U1()])
bo = cytnx.Bond(cytnx.BD_OUT, [cytnx.Qs(+1)>>1, cytnx.Qs(-1)>>1],\
                [cytnx.Symmetry.U1()])
# Initialize U(1) symmetric UniTensor
H_sym = cytnx.UniTensor([bi,bi,bo,bo]).set_name("H symmetric")\
                        .relabels(["i1","i2","j1","j2"])
# Copy data from the non-symmetric UniTensor H
H_sym.convert_from(H)
print(H_sym) # Check the blocks
```

Listing 44: Convert a non-symmetric `UniTensor` to a symmetric `UniTensor`.

Similarly, we can directly create a symmetric `UniTensor`, which represents the previous Hamiltonian. This time, the $\sigma$-matrices need to be symmetric tensors which carry quantum numbers. The $\sigma^+$ and $\sigma^-$ operators change the quantum number locally. In this case, we can see them as creation and annihilation operators, which change the particle number. Thus, the

corresponding tensors would naively not conserve the quantum number. However, we can attach an additional bond with only one index to these $\sigma$-matrices, whose quantum number corresponds to the local particle number change (`bqo` or `bqi` in Listing 45). We observe that the terms of the Hamiltonian only contain pairs which involve one $\sigma^+$ and one $\sigma^-$ operator. The Hamiltonian thus conserves the particle number, which makes it a good quantum number. We can use this to make sense of the previously introduced quantum number changing bonds: each $\sigma^+$ tensor needs to have such an additional bond, which is contracted with the additional bond of a $\sigma^-$ tensor according to the terms of the Hamiltonian. The following code example demonstrates this:

```python
# Supposes H_sym is a symmetric UniTensor that is already created
# as in Listing 44
# Create bonds with U(1) symmetry
bi = cytnx.Bond(cytnx.BD_IN, [cytnx.Qs(+1)>>1, cytnx.Qs(-1)>>1],\
                [cytnx.Symmetry.U1()])
bo = cytnx.Bond(cytnx.BD_OUT, [cytnx.Qs(+1)>>1, cytnx.Qs(-1)>>1],\
                [cytnx.Symmetry.U1()])
# Bond between sigma^+ and sigma^- needs to carry a quantum number
bqo = cytnx.Bond(cytnx.BD_OUT, [cytnx.Qs(+2)>>1],\
                [cytnx.Symmetry.U1()])
bqi = cytnx.Bond(cytnx.BD_IN, [cytnx.Qs(+2)>>1],\
                [cytnx.Symmetry.U1()])
# Create sigma matrices as symmetric UniTensors
Sp = cytnx.UniTensor([bi,bo,bqo])    # sigma^+
Sp.at([0, 1, 0]).value = +1.
Sm = cytnx.UniTensor([bqi,bi,bo])    # sigma^-
Sm.at([0, 1, 0]).value = +1.
Sz = cytnx.UniTensor([bi,bo])        # sigma^z
Sz.at([0, 0]).value = +1.
Sz.at([1, 1]).value = -1.
# Create two copies of all sigma matrices for the two sites
Sp1 = Sp.clone().set_name("Sp1").relabels(["i1","j1","q"])
Sp2 = Sp.clone().set_name("Sp2").relabels(["i2","j2","q"])
Sm1 = Sm.clone().set_name("Sm1").relabels(["q","i1","j1"])
Sm2 = Sm.clone().set_name("Sm2").relabels(["q","i2","j2"])
Sz1 = Sz.clone().set_name("Sz1").relabels(["i1","j1"])
Sz2 = Sz.clone().set_name("Sz2").relabels(["i2","j2"])
# Product of two sigma matrices at different sites;
# labels "q" are summed over
Sp1Sm2 = cytnx.Contract(Sp1, Sm2).set_name("Sp1Sm2")
Sp1Sm2.permute_(["i1","i2","j1","j2"])
Sm1Sp2 = cytnx.Contract(Sm1, Sp2).set_name("Sm1Sp2")
Sm1Sp2.permute_(["i1","i2","j1","j2"])
Sz1Sz2 = cytnx.Contract(Sz1, Sz2).set_name("Sz1Sz2")
Sz1Sz2.permute_(["i1","i2","j1","j2"])
# Create Hamiltonian as symmetric UniTensor
H2_sym = Sp1Sm2 + Sm1Sp2 + Sz1Sz2
H2_sym.set_name("H2 symmetric").relabels_(["i1","i2","j1","j2"])
# Check the blocks
print(H2_sym - H_sym) # Output: all elements are zero
```

Listing 45: Create a two-site Hamiltonian $H = \sigma_1^+ \sigma_2^- + \sigma_1^- \sigma_2^+ + \sigma_1^z \sigma_2^z$ as symmetric UniTensor.

Finally, we can convert the symmetric `UniTensor` into a non-symmetric one and compare to our previous initialization of the Hamiltonian:

```python
# Supposes H2_sym is a symmetric UniTensor that is already created
# as in Listing 45
H2=cytnx.UniTensor.zeros([2,2,2,2]).relabels((["i1","i2","j1","j2"]))\
                          .set_name("H2")
```

```
H2.convert_from(H2_sym)
# Print the Hamiltonian in matrix form
print(H2.reshape(4,4)) # Output: same data as in  Listing 43
```

Listing 46: Convert a symmetric <code>UniTensor</code> to a non-symmetric <code>UniTensor</code>.

The conversion between symmetric and non-symmetric can be very useful, for example for conveniently checking the elements of a symmetric `UniTensor` in matrix form. We note, however, that performing operations on non-symmetric tensors and then converting them to symmetric ones can lead to problems. This can, for example, happen, when linear algebra routines are performed on non-symmetric tensors. Numerical errors or simply the fact that symmetry violating elements are not explicitly forbidden can lead to non-zero elements in the wrong blocks. Even when these are ignored using the argument `force=True`, errors can accumulate in tensor network algorithms. We therefore suggest to avoid `convert_from` when possible and only use it with care. Symmetric tensors should be used wherever symmetries shall be preserved. Cytnx provides linear algebra routines that preserve the symmetries of the tensors. If operations are needed that are not natively supported by Cytnx, these should be operating on the level of the blocks to make sure symmetry violating elements can not arise.

Besides avoiding numerical errors, symmetric tensors also help in the code development by ensuring that only operations are allowed which respect the symmetries. For example, the $\sigma^+$ and $\sigma^-$ operators in Listing 45 can only come in pairs, such that their additional indices are contracted. Otherwise, open indices arise that still need to be contracted with suitable operators. This behavior is much less error prone than using non-symmetric tensors which allow for any operation, including symmetry violating ones.

## 7    Tensor contraction

All tensor network algorithms include tensor contractions, where we multiply tensors and sum over shared indices. Cytnx provides three different ways to perform these contractions. A simple way to contract indices is provided by `Contract` and `Contracts`. These functions contract two or more `UniTensor` objects by summing over all indices that have the same label on two of the tensors. We also support the API of `ncon` which allows users to contract tensors by defining the connectivity and contraction order of the bonds. We refer to the User Guide [39] for the detail. Since the user needs to specify the bonds by their indices instead of their labels, so the index order matters. This can be error prone when users have to keep track of the change of the index order in complicated tensor network algorithms. In Cytnx we introduce an advanced method for tensor network contractions via a `Network` object. It is particularly useful for contractions involving many tensors and reusable contraction diagrams. The `Network` based contraction will be explained in more details in Sec. 8.

### 7.1    Contract

Two tensors can be contracted with `Contract` as shown in the code example in Listing 47. Using this function, common indices of the two tensors are summed over. A new `UniTensor` is created to represent the result, while the input tensors are not modified. The labels of the output tensor are inherited from the input tensors. In order to avoid potential mistakes, Cytnx checks that the dimensions of the contracted indices are the same before running a contraction. For symmetric tensors, Cytnx also checks the bond direction, the quantum numbers, and the dimensions of each symmetry block.

In some scenario, one may need to change the labels of the input tensors for the contraction, while the labels of the original tensors shall stay unchanged. In this case, one can use

`UniTensor.relabel` to obtain a `UniTensor` with new labels but without actually copying the data (see Sec. 5.2). Alternatively, a `Network` can be used where no initial relabeling is needed, see Sec. 8.

```
# Define two UniTensor A and B
A = cytnx.UniTensor.ones([2,3,5]).relabels(["i","j","l"]).set_name("A")
B = cytnx.UniTensor.ones([3,1,5,4]).relabels(["j","k","l","m"])\
                        .set_name("B")
# Contract A and B by their common indices and store the result in AB
# Indices "j" and "l" are contracted
# Output tensor has indices "i", "k", and "m"
AB = cytnx.Contract(A, B)
AB.set_name("AB")
# One can check the result with print_diagram
A.print_diagram()
B.print_diagram()
AB.print_diagram()
```

Listing 47: Contract two tensors with `Contract`.

## 7.2 Contracts

`Contracts` provides the same functionality as `Contract` but allows users to contract multiple tensors. The first argument of this function is `TNs`, which is a list containing all `UniTensors` to be contracted. There are two further, optional arguments: `order` (string), which specifies a desired contraction order, and `optimal` (`True` or `False`; default `True`), for specifying if one wants to use an automatically optimized contraction order. Since, the contraction order is given in terms of the names of the `UniTensors`, the names have to be set for all tensors involved the contraction; otherwise a run-time error is raised. If `optimal=True`, a contraction order is computed in every function call, which creates some unnecessary overhead if a similar contraction is executed several times. In these cases, we recommend using a `Network` object for the contraction (see Sec. 8), where the optimal order can be calculated in the first run and be reused for consecutive contractions. When a specific order is given, the `optimal` argument should be set to `False`.

As an example, consider a contraction consisting of tensors `A1`, `A2` and `M` as sketched in Fig. 10. The corresponding code example is shown in Listing 48.

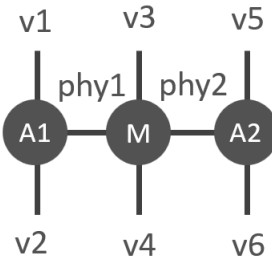

Figure 10: Diagram for contraction of three tensors.

```
# Create tensors A1, A2, M
A1 = cytnx.UniTensor.ones([2,8,8]).relabels(["phy1","v1","v2"])\
                        .set_name("A1")
A2 = cytnx.UniTensor.ones([2,8,8]).relabels(["phy2","v5","v6"])\
                        .set_name("A2")
M = cytnx.UniTensor.ones([2,2,4,4]).relabels(["phy1","phy2","v3","v4"])\
```

```
                              .set_name("M")
# Contract tensors using the auto-optimized contraction order
res = cytnx.Contracts([A1,M,A2]).set_name("res")
# Contract tensors using a user-defined contraction order
res2 = cytnx.Contracts([A1,M,A2], order="(M,(A1,A2))", optimal=False)
res2.set_name("res2")
# Check result
res.print_diagram()
```

Listing 48: Contract three tensors according to Fig. 10 with `Contracts`.

# 8 Network

Cytnx makes it easy to design, implement, and optimize the contraction of sophisticated tensor networks via the `Network` object, which contains the blueprint of the corresponding tensor network. A `Network` can be created from a string or a plain text file, which contains the description of the network in a human readable format. The `Network` can be displayed as well to check the implementation. One can contract multiple tensors at once with a `Network`. Furthermore, the contraction order can either be specified by the user, or an optimal order can be generated automatically. A `Network` is particularly useful when the same kind of tensor network contraction has to be performed repeatedly, but with different tensor objects each time. We can define the `Network` once and reuse it for several contractions. The contraction order can be optimized once after the first initialization with tensors. In the proceeding steps, this optimized order can be reused.

Using `Network` to perform a tensor contraction includes three steps. First, we need to define the target tensor network and create a `Network` object. Second, we need to load (`put`) the actual `UniTensor`s that are to be contracted. Finally, we launch the contraction. In the following we explain these steps in more detail.

## 8.1 Creating a Network

The creation of a `Network` object is demonstrated with an example here: we use a `Network` to perform two matrix-matrix multiplications for three rank-2 tensors $M_1$, $M_2$, and $M_3$ at once. The outcome is also a rank-2 tensor. The network diagram and its output tensor are shown in Fig. 11. The code example in Listing 49 shows how to create a corresponding `Network` object from a string. One can print the network to check the structure.

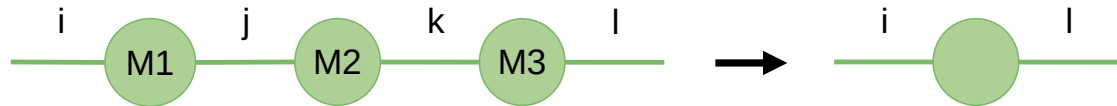

Figure 11: Network diagram for the the three matrices multiplication.

```
# Define a Network object for the contraction task in Fig. 11
net = cytnx.Network()
net.FromString(["M1: i, j",\
                "M2: j, k",\
                "M3: k, l",\
                "TOUT: i; l",\
                "ORDER: ((M1,M2),M3)"])
print(net)
```

Listing 49: Create a Network for the multiplication of three matrices according to Fig. 11.

The contraction is defined by the rule that all the common indices are summed over, and the remaining indices will be the indices of the output tensor. The keyword `TOUT` is used to specify the *order* of the remaining indices that are not summed over. In this example, indices labeled by `i` and `j` are not contracted and they become the first and second index of the output tensor. Consequently, if the output tensor represents a scalar (rank-0 tensor), no labels should be given after `TOUT:`. The keyword `ORDER` is used to specify a user-defined contraction order. If nothing is specified after `ORDER:`, the default order is contracting the tensors in the order of their appearance in the creation of the `Network`. One can modify the `ORDER` after the creation of the `Network` or let Cytnx find an optimal contraction order after all tensors are loaded, see Sec. 8.3.

One can also create a `Network` object from a file. For this, the same content as in the afore-mentioned network creation string (Listing 50) is written to a plain text file. Here, we name the file `example.net`. Such a network file can then be loaded with the `Network` initializer or by using `Network.Fromfile`. One can also write a `Network` object to a network file using `Network.Savefile`. The code example below (Listing 51) illustrates loading and saving a network file.

```
M1: i, j
M2: j, k
M3: k, l
TOUT: i; l
ORDER: ((M1,M2),M3)
```

Listing 50: Network file "example.net", corresponding to the network in Listing 49.

```
# Create a Network object from the network file in Listing 50
net = cytnx.Network("example.net")
# Write the current network to a network file
net.Savefile("savefile")
# Read a network file
net.Fromfile("savefile.net")
```

Listing 51: Create a Network from a file.

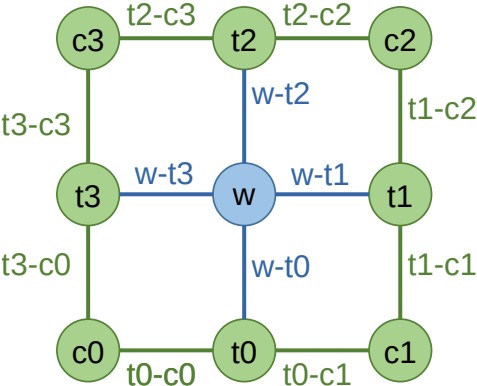

Figure 12: Network diagram for the corner transfer matrix algorithm.

To further demonstrate the power of `Network`, we consider a contraction that appears in the corner transfer matrix algorithm [44, 45]. The tensor network diagram is sketched in

Fig. 12. In the code example below (Listing 52), we create a `Network` from a string for this case. The example demonstrates that it is also straightforward to translate a more complicated network diagram into a string, which defines the corresponding `Network`. The `Network` object is thus particularly useful for algorithms in higher dimensions, where many tensors need to be contracted. We note that in the case shown here, `TOUT:` is empty because the outcome tensor is a scalar.

```
net = cytnx.Network()
net.FromString(["c0: t0-c0, t3-c0",\
                "c1: t1-c1, t0-c1",\
                "c2: t2-c2, t1-c2",\
                "c3: t3-c3, t2-c3",\
                "t0: t0-c1, w-t0, t0-c0",\
                "t1: t1-c2, w-t1, t1-c1",\
                "t2: t2-c3, w-t2, t2-c2",\
                "t3: t3-c0, w-t3, t3-c3",\
                "w: w-t0, w-t1, w-t2, w-t3",\
                "TOUT:",\
                "ORDER: (((((((((c0,t0),c1),t3),w),t1),c3),t2),c2)"])
```

Listing 52: Create a `Network` for the corner transfer matrix method as shown in Fig. 12.

We emphasize that the labels used in the `Network` are dummy indices only for the network, and are completely independent of the labels in the `UniTensor` objects that are going to be contracted. This design gives us the flexibility to define a `Network` object without knowing the labeling conventions of the `UniTensor`s to be contracted. In particular, the `Network` can be loaded with different `UniTensor`s for each contraction and be reused.

## 8.2 Putting UniTensors and performing a contraction

A `Network` is a skeleton that does not contain the actual tensor objects to be contracted. To perform the contraction, one needs to specify a `UniTensor` object for each tensor defined in the `Network`, and make a connection between the dummy indices in the `Network` and the actual indices of the `UniTensor`s. This is done by using `Network.PutUniTensor`. To give an example, we consider the `Network` defined in Listing 49 and the following `UniTensor`s which shall be contracted:

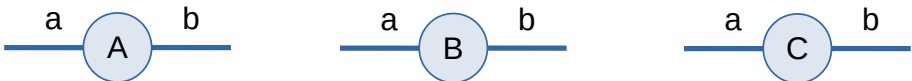

Figure 13: UniTensors to be put into the Network.

Note that the indices of `A`, `B`, and `C` have common index labels $a$ and $b$. This is often the case in certain tensor network applications. We load the `Network` with the specific `UniTensor`s by using `Network.PutUniTensor` for each tensor individually. The first argument is the name of the tensor to be specified in the network, and the second argument is the `UniTensor` object as a realization of that tensor. The third argument is a list of the labels in the `UniTensor` in the order of the indices of the `Network`. For example, `A` is a realization of `M1`, and the indices `a` and `b` in `A` correspond to the indices `i` and `j` in `M1`. Similarly, `B(C)` is a realization of `M2(M3)`, and the indices `a` and `b` in `B(C)` correspond to the indices `j(k)` and `k(l)` in `M2(M3)`. When all tensors in the network are specified, one can perform the contraction by simply calling

`Network.Launch` to obtain the result. Note that the output tensor inherits the index order and labels from the specifications following `TOUT:`.

```python
# Supposes net is a Network that is already created as in Listing 49
# Create UniTensors that are going to be contracted
A = cytnx.UniTensor.ones([2,2]).relabels(["a","b"]).set_name("A")
B = cytnx.UniTensor.ones([2,2]).relabels(["a","b"]).set_name("B")
C = cytnx.UniTensor.ones([2,2]).relabels(["a","b"]).set_name("C")
# Put UniTensors into a Network object
net.PutUniTensor("M1", A, ["a","b"])
net.PutUniTensor("M2", B, ["a","b"])
net.PutUniTensor("M3", C, ["a","b"])
# Launch the contraction
Tout = net.Launch()
```

Listing 53: Load tensors into the network and launch contraction.

## 8.3 Setting the contraction order

The efficiency of a tensor network contraction depends on the order of the contractions. This order can be specified when the `Network` is defined (see Sec. 8.1). However, one can also reset the contraction order using the method `Network.setOrder`. A certain contraction order can be forced in the same way as explained in Sec. 8.1. Alternatively, one can let Cytnx compute the optimal order by setting `Network.setOrder(optimal=True)`. Note that in this case, the optimal order will be computed (and stored) *every time* when a contraction is launched. This is necessary because the optimal order depends on the bond dimensions of the input tensors, which can be different for each contraction. In practice, one often increases the dimensions of the indices during the simulations until they are kept constant when they reach a certain value. Then, the optimal contraction order typically does not change anymore. In this cases, one can set `Network.setOrder(optimal=True)` up to some large enough bond dimensions, and then set `Network.setOrder(optimal=False)` such that the `Network` will no longer compute the optimal order before the contractions, but just use the stored contraction order. Similarly, the dimensions do not change when a network contraction is used in an iterative solver many times. Then, the contraction order only needs to be optimized in the first iteration. One can check the current contraction order using `Network.getOrder`:

```python
# Supposes net is a Network that is already created as in Listing 49
# and all tensors are loaded (put) as in Listing 53
# Calculate the optimal contraction order for the loaded tensors
net.setOrder(optimal=True)
# Set a specific contraction order
net.setOrder(optimal=False, contract_order='(M2,(M1,M3))')
# Print the current contraction order
print(net.getOrder()) # Output: (M2,(M1,M3))
```

Listing 54: Set and get the contraction order.

## 8.4 Advanced example: Expectation value of a Projected Entangled Pair State

In this advanced example, we show how a more complicated tensor network contraction can be implemented by using `Network`. Specifically, we consider the contraction for the expectation value of a two-dimensional tensor network state. Physical states in two spatial dimensions, in particular ground states, can be approximated by a Projected Entangled Pair State (PEPS) [46]. This tensor network consists of one tensor at each lattice site. One index of each tensor corresponds to the physical degrees of freedom at this site (dotted lines in Figs. 14 and 15). Further

internal bonds connect each tensor to its nearest neighbors. Fig. 14 shows a PEPS tensor in two spatial dimensions.

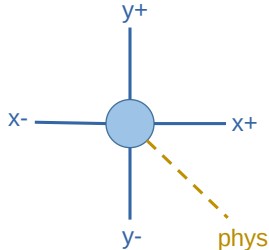

Figure 14: PEPS tensor and labeling convention.

A typical PEPS algorithms prepares a desired state, for example the ground state of a Hamiltonian. Then, expectation values are measured as the quantities of interest for physics [1]. The boundary matrix product state (boundary MPS) method is used to efficiently calculate expectation values. See Ref. [47] as an example of a finite lattice PEPS calculation with more details on the algorithm.

The expectation value of a nearest neighbor operator $op$ corresponds to the tensor network in Fig. 15. Here, the blue tensors $t0$ and $t1$ are the PEPS tensors that the operator acts on, while the starred versions are there complex conjugate counterparts. The green tensors $b0$ to $b5$ represent a boundary matrix product state, which is used to approximate the PEPS tensors at the remaining sites.

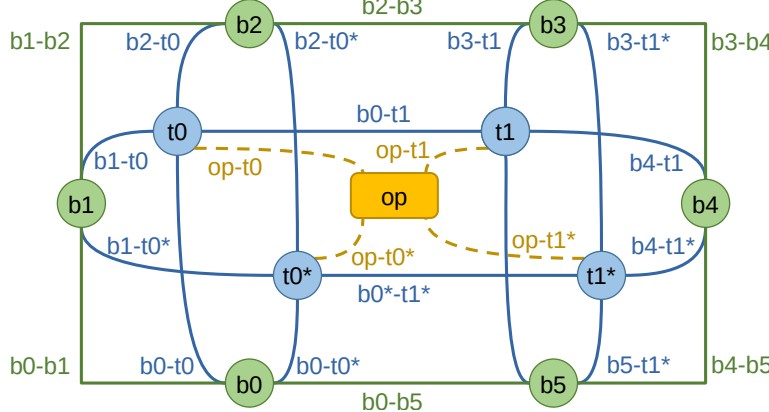

Figure 15: Expectation value for a nearest neighbor operator acting on a PEPS. The boundary MPS approximation is used. Symbology – yellow: operator; blue: PEPS tensors; starred blue: corresponding complex conjugate PEPS tensors; green: boundary matrix product state, an approximation of the remaining sites. Adapted from Ref. [48].

The diagram in Fig. 15 can directly be translated into a `Network` file. One writes the names of the tensors and their indices one by one to the file:

```
#network file peps_exp_val.net
b0:  b0-b5,  b0-b1,   b0-t0,   b0-t0*
b1:  b0-b1,  b1-b2,   b1-t0,   b1-t0*
b2:  b1-b2,  b2-b3,   b2-t0,   b2-t0*
b3:  b2-b3,  b3-b4,   b3-t1,   b3-t1*
b4:  b3-b4,  b4-b5,   b4-t1,   b4-t1*
b5:  b4-b5,  b0-b5,   b5-t1,   b5-t1*
t0:  op-t0,  t0-t1,   b0-t0,   b1-t0,   b2-t0
```

```
t0*: op-t0*, t0*-t1*, b0-t0*, b1-t0*, b2-t0*
t1:  op-t1,  t0-t1,   b3-t1,  b4-t1,  b5-t1
t1*: op-t1*, t0*-t1*, b3-t1*, b4-t1*, b5-t1*
op:  op-t0,  op-t0*,  op-t1,  op-t1*
TOUT:
ORDER:
```

Listing 55: Network file `peps_exp_val.net` describing the contraction in Fig. 15.

The actual contraction is done by loading the `Network` file and using `PutUniTensor` with the correct `UniTensors`. This way, the expectation value can be calculated for tensors at different sites, and also other naming conventions of the indices could be used without changing the format of the `Network` file. Our index labeling convention for the PEPS tensors can be seen in Fig. 14. The names of the boundary MPS tensors start with "bMPS" in our convention, and the index labels indicate to which kind of tensor the index connects to.

When loading the actual tensors, the indices of a `UniTensor` have to be assigned to the corresponding ones in the Network. This can simply be done by the index labels in the correct order as the third argument of `net.PutUniTensor`. Figure 16 shows an example for this mapping. The code to put the `UniTensors` and launch the contraction is shown below in Listing 56.

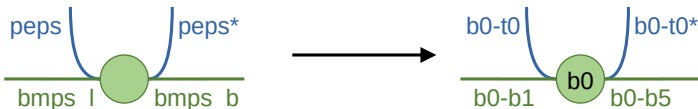

Figure 16: Example for the mapping of index labels. Left: `UniTensor bMPS_b0` with index labels. Right: Abstract tensor definition `b0` in the `Network`, see Fig. 15. The labels are matched by giving the bond labels as a third argument in `Network.PutUniTensor`as in Listing 56.

```
# Supposes "peps_exp_val.net" is a Network that is already created as
# in Listing 55 and the UniTensors to be put exist
net = cytnx.Network("peps_exp_val.net")
net.PutUniTensor("t0",  ten0,     ["phys", "x+", "y-", "x-", "y+"])
net.PutUniTensor("t0*", conjten0, ["phys", "x+", "y-", "x-", "y+"])
net.PutUniTensor("t1",  ten1,     ["phys", "x-", "y+", "x+", "y-"])
net.PutUniTensor("t1*", conjten1, ["phys", "x-", "y+", "x+", "y-"])
net.PutUniTensor("b0",  bMPS_b0,  ["bmps_b", "bmps_l", "peps", "peps*"])
net.PutUniTensor("b5",  bMPS_b1,  ["bmps_r", "bmps_b", "peps", "peps*"])
net.PutUniTensor("b2",  bMPS_t0,  ["bmps_l", "bmps_t", "peps", "peps*"])
net.PutUniTensor("b3",  bMPS_t1,  ["bmps_t", "bmps_r", "peps", "peps*"])
net.PutUniTensor("b1",  bMPS_l,   ["bmps_b", "bmps_t", "peps", "peps*"])
net.PutUniTensor("b4",  bMPS_r,   ["bmps_t", "bmps_b", "peps", "peps*"])
net.PutUniTensor("op",  op,       ["peps_left", "peps_left*",
                                   "peps_right","peps_right*"])
exp_val = net.Launch().get_elem([0])
```

Listing 56: Put UniTensors and launch contraction. The index labels of the PEPS tensors are shown in Fig. 14. The boundary MPS tensors are named according to their position t(op), b(ottom), l(eft), r(ight). Their index labels indicate the kind of tensor they connect to. See Fig. 16 for an example of the index convention and mapping.

The diagram in Fig. 15 is intended for calculating an expectation value of an operator that acts on a nearest-neighbor pair in x-direction. However, the same `Network` file can be used also for operators acting in y-direction. Only the arguments of `PutUniTensors` need to be

adapted. This way, tensor diagrams can be implemented as network files and be used flexibly, without the need to rewrite them for different tensors to be loaded. Also, the user does not need to care about the index order inside the `UniTensor` if labeled indices are used.

# 9 Tensor decomposition and linear algebra

Tensor decompositions by linear algebra algorithms are essential ingredients in tensor network applications. Cytnx supports an extensive list tensor operations and linear algebra functions and we refer to the User Guide [39] for the complete list and further documentation. Typically, linear algebra functions require matrices as input. Consequently, before calling a linear algebra function, Cytnx *internally* maps a `UniTensor` to a matrix. In Sec. 9.1 we explain how to ensure that the correct mapping is carried out. Then, we show how to perform singular value decomposition (SVD) (Sec. 9.2), eigenvalue decomposition (Sec. 9.3), and QR decomposition (Sec. 9.4) respectively.

## 9.1 Mapping from tensor to matrix

Cytnx provides several linear algebra functions and tensor decomposition methods. The library calls established and well optimized linear algebra functions, which require matrices as input. Therefore, the tensors need to be represented by matrices in any of the methods. This is done by classifying each index in a tensor to be either part of the resulting row index or column index, and then combining all the row (column) indices into one row (column) index for the matrix representation. The dimension of the final row (column) index is the product of the dimensions of all the original row (column) indices.

   The `UniTensor` objects have an attribute called `rowrank` to control the mapping from a tensor to a matrix. In matrix representation, the first `rowrank` indices are treated as row indices, and the remaining indices are treated as column indices. There are different ways of mapping a tensor to a matrix depending on which indices are selected as row or column indices. Before calling a tensor decomposition method, one needs to prepare the tensor such that it has the desired mapping. This is done by 1) permuting the indices such that all the row indices are in front of all the column indices, and 2) setting `rowrank` to the number of row indices by using `UniTensor.set_rowrank` or `UniTensor.set_rowrank_`.

   The following example shows how to prepare a specific matrix representation of a rank-3 `UniTensor`, where one of the indices is interpreted as the row index and the other two indices form the column index:

```
# Create a rank-3 tensor with shape [2,2,6]
M = cytnx.UniTensor.arange(2*2*6).reshape(2,2,6)\
                            .relabels(["a","b","c"]).set_name("M")
# 1. Permute the indices to ("c","a","b") ordering,
M.permute_(["c","a","b"])
# 2. Set rowrank=1, such that "c" is the row index,
# and "a", "b" are the column indices.
M.set_rowrank_(1)
print(M.rowrank()) # Output: 1
M.print_diagram()

''' -------- Output ---------
tensor Name : M
tensor Rank : 3
block_form  : False
is_diag     : False
on device   : cytnx device: CPU
```

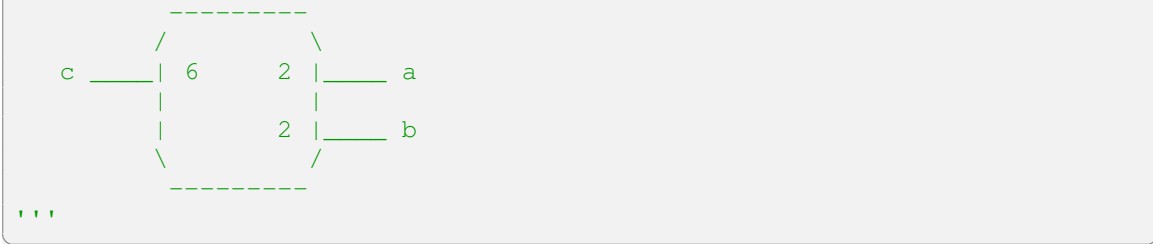

```
              ---------
             /         \
   c _____| 6      2 |_____ a
             |         |
             |       2 |_____ b
             \         /
              ---------
'''
```

Listing 57: Matrix form of a `UniTensor`

We see that `M.rowrank` return the `rowrank` information, which is also encoded in the diagram created by `UniTensor.print_diagram` – the indices at the left (right) side are the row (column) indices.

## 9.2 Singular value decomposition

**Without truncation.** The singular value decomposition (SVD) decomposes a matrix $M$ into the product of three matrices $U$, $S$, and $V^\dagger$ (i.e., $M = USV^\dagger$), where $S$ is a diagonal matrix and its non-negative elements are called singular values. $U$ and $V$ satisfy $U^\dagger U = I$ and $V^\dagger V = I$ with $I$ the identity matrix. In Cytnx, one can perform a SVD by using the function `linalg.Svd`. There is an optional argument `is_UvT` (default `True`): If `is_UvT=True`, the function returns three tensors $S$, $U$, and $V^\dagger$; if `is_UvT=False`, the function returns only the singular-value tensor $S$. Note that this function does *not* perform any truncation. We also note that $U$ inherits the row labels of $M$, while $V^\dagger$ inherits the column labels of $M$. The other labels are automatically generated in such a way that one can use `Contracts` to perform the matrix multiplication of $U$, $S$, and $V^\dagger$. However, the indices after the contraction with `Contracts` might have a different order and need to be permuted before comparing to the original tensor $M$. The following example shows how to perform a SVD of the previously created rank-3 tensor:

```
# Supposes M is a UniTensor that is already created as in Listing 57
# Perform SVD on M
S, U, Vdag = cytnx.linalg.Svd(M)
# Compute the product of U*S*Vdag
Msvd = cytnx.Contracts([U,S,Vdag])
Msvd.permute_(M.labels())
# Check that M = U*S*Vdag
Diff = M - Msvd
print(Diff.Norm()) # Output: 2.79115e-14
```

Listing 58: Singular value decomposition without truncation.

**With truncation.** In practice, one often uses a SVD as an approximating method by truncating the linear space based on the singular values. A SVD with truncation can be executed by calling `linalg.Svd_truncate`. This function performs a full SVD of the initial tensors, and then truncates it to keep only the largest singular values. `linalg.Svd_truncate` has several additional arguments:

- `keepdim` specifies the maximal dimension to be kept after the truncation.

- `err` defines the threshold for singular values to be kept; the linear space with singular value smaller than `err` will be truncated. Note that the singular values are not normalized before the truncation.

- return_err; for return_err=1, the function returns an additional one-element `UniTensor` that stores the largest of the truncated singular values; for return_err=2, a `UniTensor` containing all the truncated singular values is returned instead; for return_err=0 (default), no truncation error is returned

Here is an example of an SVD with truncation:

```
# Supposes M is a UniTensor that is already created as in Listing 57
# Perform SVD without truncation and return only the singular values
S_full = cytnx.linalg.Svd (M, is_UvT=False)
# Perform SVD with truncation
S_trunc, U, Vdag, s_err = cytnx.linalg.Svd_truncate\
                    (M, keepdim=3, err=1e-10, return_err=1)
# Compare the full sigular values with the kept and the truncated ones
print(S_full)   # [6.56235e+01 4.18988e+00 2.92323e-15 8.22506e-16 ]
print(S_trunc)  # [6.56235e+01 4.18988e+00 ]
print(s_err)    # [2.92323e-15 ]
```

Listing 59: Singular value decomposition with truncation.

## 9.3 Eigenvalue decomposition

The eigenvalue decomposition, also known as matrix diagonalization, is defined for square matrices only. It decomposes a square matrix $M$ to $VDV^{-1}$, where $D$ is a diagonal matrix and its elements are called the eigenvalues. The columns of $V$ correspond to the right eivenvectors of $M$. If $M$ is a Hermitian matrix, all the eigenvalues must be real, and $V$ becomes a unitary matrix satisfying $V^\dagger V = VV^\dagger = I$. Consequently $M = VDV^\dagger$ for a Hermitian matrix.

One can perform the eigenvalue decomposition by using `linalg.Eig` for a generic square matrix or `linalg.Eigh` for a Hermitian matrix. Since the input matrix must be a square matrix, the row and the column dimensions in the matrix representation must be equal; otherwise, an error is raised. One can use the optional argument `is_V` (default `True`) to determine if the eigenvector matrix $V$ shall be returned. A complete description of the optional arguments can be found in the User Guide [30]. Here is an example of an eigenvalue decomposition of a Hermitian matrix:

```
# Create a randomly initialized Hermitian matrix
uT = cytnx.UniTensor.uniform([4,4], low=-1., high=1.)
uT = uT + uT.Conj().Transpose()
uT = uT.relabels(["a","b"]).set_name("uT").set_rowrank(1)
# Eigenvalue decomposition
eigvals, V = cytnx.linalg.Eigh(uT)
Vdag = V.Conj().Transpose()
# Set labels
V.relabels_(["a","_aux_L"]).set_name("V")
eigvals.relabels_(["_aux_L","_aux_R"]).set_name("eigvals")
Vdag.relabels_(["_aux_R","b"]).set_name("Vdag")
# Compare uT with V*eigvals*Vdag
Diff = uT - cytnx.Contracts([V,eigvals,Vdag])
print(Diff.Norm()) # Output: 1.59153e-15
# Create identity matrix
eye = cytnx.UniTensor.zeros([4,4])
for i in range(4):
    eye[i,i] = 1
eye.relabels_(["_aux_L","_aux_R"]).set_name("eye")
# Compare eye with V*eye*Vdag
Diff = eye - cytnx.Contracts([V,eye,Vdag])
print(Diff.Norm()) # Output: 6.09676e-16
```

Listing 60: Eigenvalue decomposition.

If one is only interested in the lowest few eigenvalues, it can be very inefficient to first execute the contraction of a tensor network to obtain a matrix and then pass this matrix to the eigenvalue decomposition function. Methods like the Lancsoz solver for finding the smallest eigenvalues only require a linear operator, which defines how a vector is transformed. This can be a matrix multiplication, but also a more complicated tensor network contraction or any general linear transformation of the input vector. Cytnx provides a way to define such an operator as a `LinOp` object and pass it to the `Lanczos` linear algebra function to find the lowest few eigenvalues and corresponding eigenvectors of a Hermitian linear operator. More details and example code can be found in the User Guide [49].

## 9.4 QR decomposition

The QR decomposition decomposes a matrix $M$ to the form $M = QR$, where $Q$ is a column-orthogonal matrix ($Q^T Q = I$), and $R$ is an upper-right triangular matrix. One can perform a QR decomposition by using `cytnx.linalg.Qr`:

```
# Initial matrix
uT = cytnx.UniTensor.arange(5*4).reshape(5,4).relabels(["a","d"])\
                            .set_name("uT").set_rowrank(1)
# QR decomposition
Q, R = cytnx.linalg.Qr(uT)
# Compare uT with Q*R
Diff = uT - cytnx.Contract(Q,R)
print(Diff.Norm()) # Output: 1.14885e-14
# Check properties of Q,R
print(R) # upper triangular
QT = Q.Transpose().relabels_(["a","b"])
Q.relabels_(["b","c"])
eye = cytnx.UniTensor.zeros([4,4])
for i in range(4):
    eye[i,i] = 1
Diff = cytnx.Contract(QT, Q) - eye
print(Diff.Norm()) # Output: 6.13418e-16
```

Listing 61: QR decomposition.

# 10 Device and GPU support

In Cytnx, the tensor elements can either be stored in the memory accessible to CPUs or GPUs. The use of GPUs can speed up tensor contractions and therefore shorten the run-time of tensor network algorithms. Using GPUs is very simple in Cytnx: After creating a tensor directly on a GPU or moving a tensor from CPU to GPU, the functions and methods provided by Cytnx can be used without any changes of the API. Therefore, tensor network algorithms can easily be ported to GPUs by only changing the initialization step of all tensors. For example:

```
# Create two UniTensors on a GPU
A = cytnx.UniTensor.ones([2,2,2], device=cytnx.Device.cuda+0,\
                        name="A", labels=["alpha","beta","gamma"])
B = cytnx.UniTensor.ones([2,2], device=cytnx.Device.cuda+0,\
                        name="B", labels=["beta","delta"])
# Or, using .to
A = cytnx.UniTensor.ones([2,2,2]).to(cytnx.Device.cuda+0)\
                    .set_name("A").relabels(["alpha","beta","gamma"])
B = cytnx.UniTensor.ones([2,2]).to(cytnx.Device.cuda+0)\
                        .set_name("A").relabels(["beta","delta"])
# Contract the two UniTensors with the same API as for CPU
```

```
AB = cytnx.Contract(A, B)
```
Listing 62: Create and contract two tensors on a GPU.

Since tensor elements can be stored in the memory of a given CPU or GPU on multi-CPU/GPU systems, it is important that the user ensures tensors are on the same device when they are used together in contractions or linear algebra functions. In Cytnx, the object `cytnx.Device` handles all device properties. The `Device` of a `UniTensor` can be seen when printing the tensor by `print_diagram` or `print` (see Sec. 5.4).

## 10.1  Device status

**Number of threads.**    To see how many threads can be used in the current program by Cytnx, one can check `Device.Ncpus`.

```
print(cytnx.Device.Ncpus)
```

**Number of GPUs.**  The number of GPUs available to Cytnx in the current program can be checked with `Device.Ngpus`.

```
print(cytnx.Device.Ngpus)
```

If Cytnx is not compiled with CUDA available, `Device.Ngpus` returns `0`.

## 10.2  Initializing tensors on CPU and GPU

For the GPU computations, Cytnx uses the Magma [50] library as well as several CUDA based third-party libraries like cuBlas [51], cuSolver [52], cuTensor [53] and cuQuantum [54]. The performance of tensor network algorithms implemented in Cytnx can be significantly accelerated when using GPUs, as shown in Sec. 11. Developers of tensor network algorithms can directly benefit from the GPU acceleration.

Users can easily transfer tensor network algorithms implemented in Cytnx from the CPU to GPU version with minimal code modifications. Cytnx provides two ways to create objects on GPU devices: either by initializing objects on a GPU directly or by using the `to()` method to convert objects stored in the CPU memory.

**Initializing tensors on a GPU.**   `UniTensor` objects with data stored in the GPU memory can be created with the argument `device` in the initialization step. For example:

```
# Create a UniTensor on CPU; initialized with zeros
uT = cytnx.UniTensor.zeros([3,4,5], device=cytnx.Device.cpu)\
                .set_name("UniTensor").relabels(["a","b","c"])
# Create a UniTensor on GPU; initialized with zeros
uTgpu = cytnx.UniTensor.zeros([3,4,5], device=cytnx.Device.cuda+0)\
                .set_name("UniTensor on GPU").relabels(["a","b","c"])
# Check device
uT.print_diagram()       # on CPU
uTgpu.print_diagram()    # on GPU
```
Listing 63: Create a `UniTensor` on a CPU or GPU.

Other initialization methods like `ones`, `arange`, `ones`, `uniform`, `normal`, or `eye` can be used similarly.

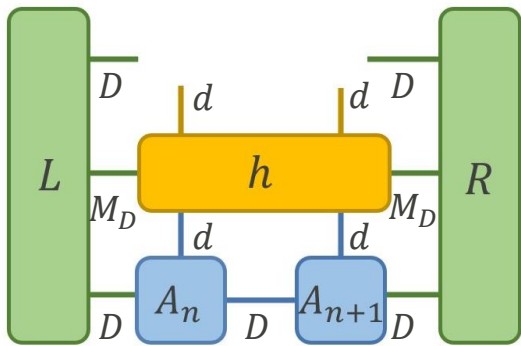

Figure 17: The tensor contraction which is used to update the $A_n$ and $A_{n+1}$ tensors during a DMRG sweep. It involves the effective Hamiltonian composed of the left and right environment tensors ($L$ and $R$) and the two-site Hamiltonian tensor ($h$). The effective Hamiltonian is contracted with the effective wavefunction composed of the MPS tensors $A_n$ and $A_{n+1}$ at sites $n$ and $n+1$. The dimension of each index is indicated besides the bond, where $d$, $D$ and $M_D$ are the physical dimension, bond dimension and MPO bond dimension respectively. In our benchmark of the XX model: $d = 2$, $M_D = 4$ and $D$ is varied.

**Data Transfer between Devices.** The library provides a convenient way to move the data of Cytnx objects between CPU and GPU with the `to` or `to_` methods. For example, we create a `UniTensor` in the memory accessible by the CPU and transfer it to the GPU with gpu-id=0:

```
uT = cytnx.UniTensor.ones([2,2]).set_name("uT").relabels(["a","b"])
uTgpu = uT.to(cytnx.Device.cuda+0).set_name("uTgpu")
uT.print_diagram()      # on CPU
uTgpu.print_diagram()   # on GPU
uT.to_(cytnx.Device.cuda)
uT.print_diagram()      # on GPU
```

Listing 64: Transfer a `UniTensor` between CPU and GPU.

If only specific parts of an algorithm are suitable for GPU simulation, it is recommended to use the `to` method to transfer the objects between CPU and GPU.

## 11 Benchmarks

To test the performance of Cytnx, we compare the execution times of two typical tensor operations in Cytnx and ITensor [31] on a CPU and on a GPU. ITensor is implemented in both Julia and C++. In this comparison, we use the C++ version of ITensor (v3.0.0). One of the main applications of tensor networks is to use them as a variational ansatz for a quantum many-body state. Here we benchmark the performance on a variational method, the density matrix renormalization group (DMRG) [2, 55], which computes the ground state of a given Hamiltonian within a tensor network approximation. In particular, we consider the one-dimensional XX model with $N$ spins:

$$\hat{H} = \sum_{j=1}^{N-1} (\hat{S}_j^x \hat{S}_{j+1}^x + \hat{S}_j^y \hat{S}_{j+1}^y).$$

The Hamiltonian conserves the total spin $\sum_{j=1}^{N} S_j^z$ in the $z$ direction, which can be used as a quantum number. In a DMRG algorithm, one represents the ground state as a network of

tensors $A_i$, where $i = 1, \cdots N$. These tensors are updated sequentially from one boundary to the other in a DMRG sweep. An essential step in the DMRG sweep is the update of two tensors $A_n$ and $A_{n+1}$, where a tensor contraction as sketched in Fig. 17 is needed. In the figure, $d$, $D$, and $M_D$ correspond to the physical dimension, bond dimension and MPO bond dimension respectively. Since the bond dimension $D$ is related to the accuracy of the DMRG algorithm, we use $D$ as the control parameter in the benchmark. It is expected that the computational time scales as $D^3$ for large $D$. An implementation of DMRG using Cytnx can be found in the examples section in Cytnx's User Guide [30].

**CPU Benchmark.** We first benchmark the performance on an Intel(R) Core i7-9700 CPU. We perform the test with symmetric tensors which preserve the total spin, as well as unconstrained tensors without explicitly implemented symmetries. In Fig. 18 (a), we show comparisons of wall times for the tensor contraction sketched in Fig. 17, as a function of $D$ on a log-log scale with Cytnx and ITensor. One can see that the performance of the two libraries is very similar. As expected, using symmetric tensors can reduce the computational costs except for very small bond dimensions. The parallelization with 8 threads reduces the execution time for the contraction compared to the single threaded run. Fig. 18 (b) shows similar comparisons but for a whole DMRG sweep. Again, Cytnx and ITensor have a similar performance. We mention that the DMRG algorithm is implemented in slightly different ways in the two libraries, and additional observables are measured in ITensor. Therefore, it can only be concluded that the run-times of Cytnx are comparable to those of ITensor, even though one of the libraries might be slightly faster for certain tasks.

**GPU Benchmark.** We next benchmark the performance of Cyntx and ITensor on GPU. The tests are executed on a machine with an AMD EPYC 7742 processor and a NVIDIA A100 GPU with 40GB memory. For the CPU runs, the parallelization is through the Intel MKL with 128 threads. Similar to the CPU benchmark, we test the performance of (a) the tensor contraction sketched in Fig. 17 and (b) the DMRG sweep. The benchmark results are shown in Fig. 19(a) and (b) respectively. In these benchmarks, one can observe a significant speedup for large bond dimensions when a GPU is used. The execution times for the tensor contraction show a 11× speedup can be reached for large bond dimensions. For a single iteration of an DMRG update, we gain a 2-3× speedup for large bond dimensions.

Summarizing, we conclude that Cytnx has a similar performance as ITensor. Cytnx allows users to run the most computationally expensive tasks on a GPU without much changes in the code by the user. This can reduce the computational time significantly.

## 12 Summary

We presented a library designed for high-dimensional tensor network algorithms. This library provides an easy-to-learn framework for new users to start developing their first tensor network code in a short period of time, yet comes with all the bells and whistles for advanced algorithms. This way, applications can quickly and conveniently be implemented and tested, but also be improved and optimized in a later step if desired. By design, the Cytnx Python and C++ interfaces follow the conventions in popular libraries for an intuitive usage. With almost identical APIs, users can easily switch between Python and C++ codes. As a benchmark, we compare typical tensor network operations against ITensor and obtain similar performance. Following a similiar syntax as in PyTorch, users can benefit from GPU acceleration with only minimal changes of their code. In our benchmark, a significant boost of the performance can be achieved when tensor operations are executed on a GPU.

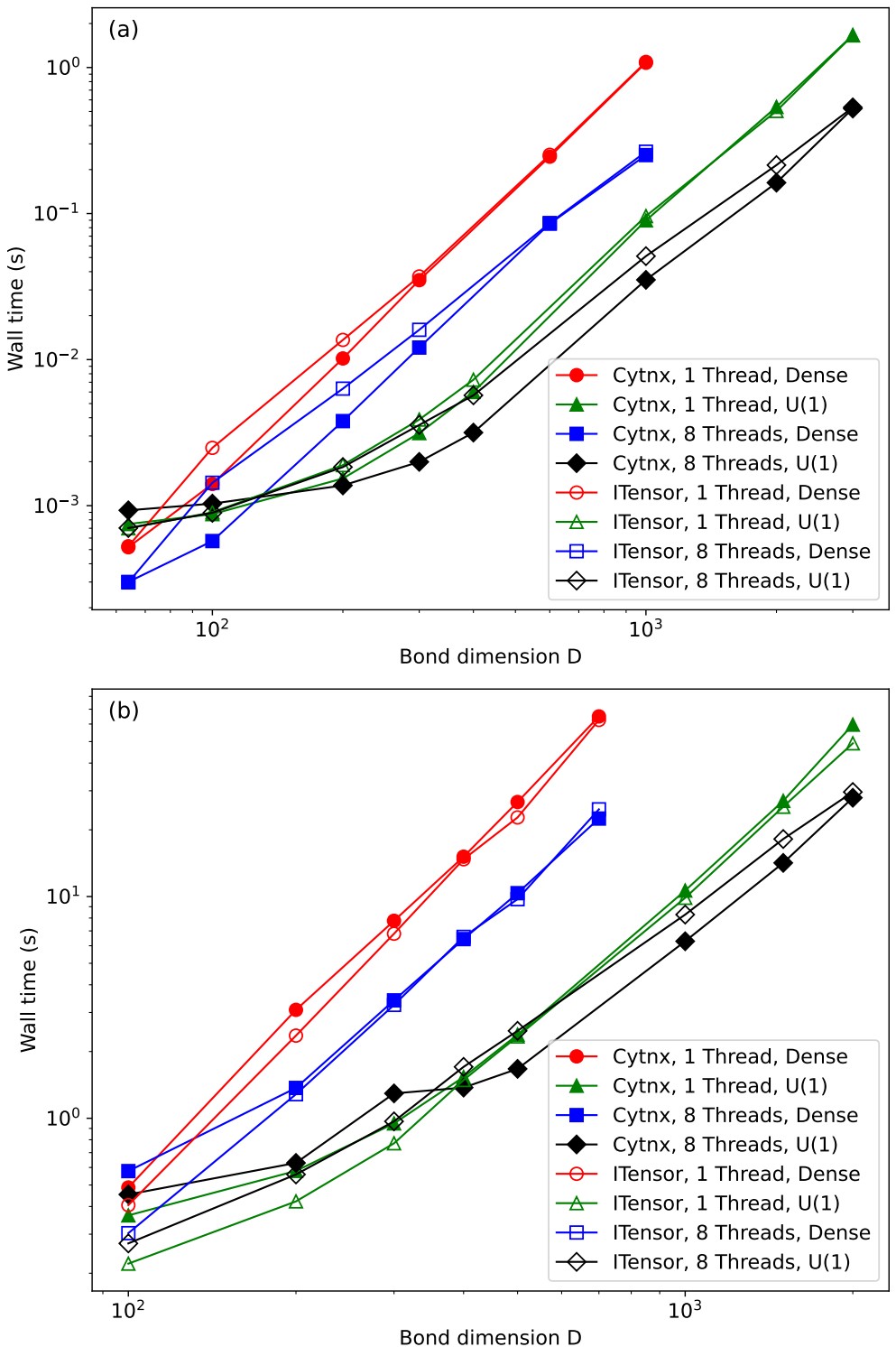

Figure 18: Computational time (wall time) for (a) the tensor contraction sketched in Fig. 17 and (b) the DMRG sweep as a function of the bond dimensions $D$. Comparison between Cytnx and ITensor for dense and symmetric tensors, performed using one or eight threads on an Intel(R) Core i7-9700 CPU.

(a)

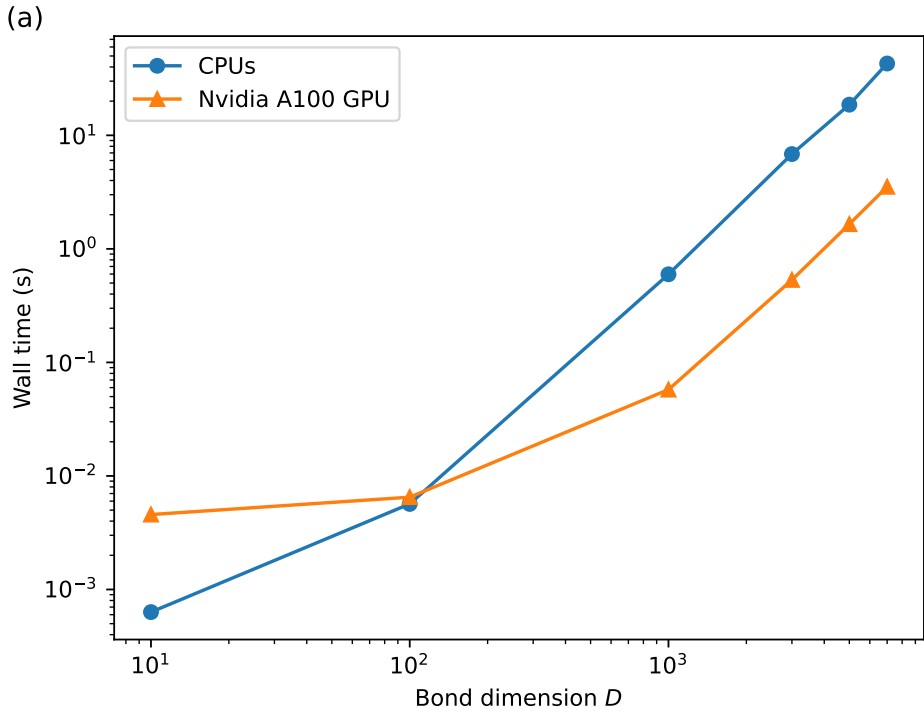

(b)

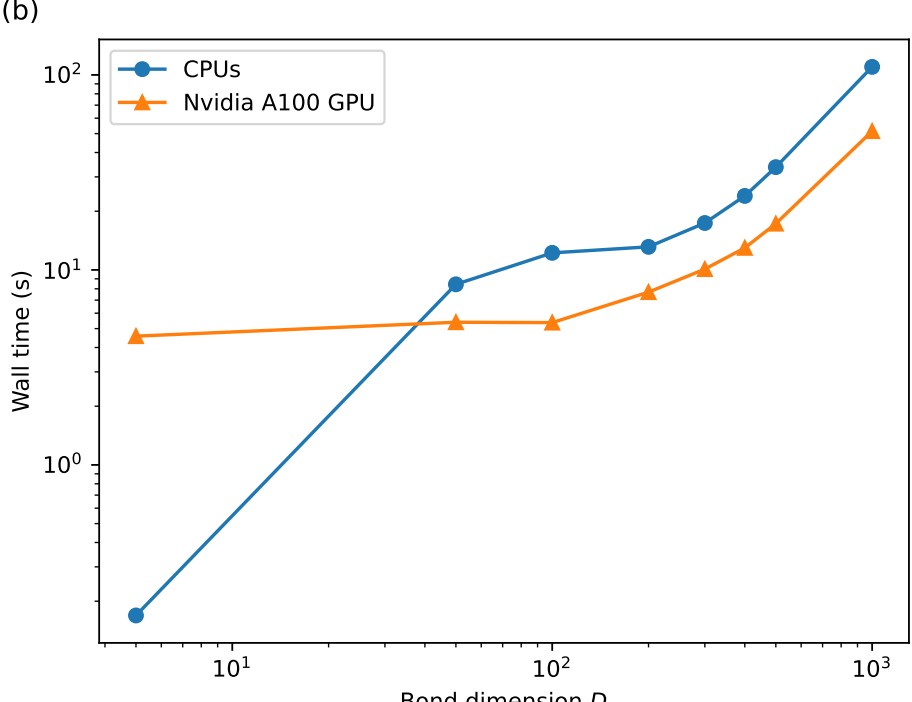

Figure 19: Computational time (wall time) for (a) the tensor network contraction sketched in Fig. 17 and (b) a DMRG sweep with several bond dimensions of the matrix product state. Comparison between CPU and GPU calculations; execution times on an AMD EPYC 7742 CPU with 128 threads and a NVIDIA A100 (40GB) GPU.

There are several future directions for Cytnx. On the technical side, tensor network algorithms are often described as tensor diagrams, therefore a GUI interface that can automatically generate code from a tensor diagram, similar to TensorTrace [56], will be helpful. Also, automatic differentiation has become an important tool for tensor network computations [57–60]. In the near future, Cytnx will support automatic differentiation through a PyTorch backend.

Many physics problems involve fermions. In these cases, exchanging the order of two particles results in a sign flip because of the anti-commutation relation between fermionic operators. In tensor network algorithms, this can be implemented with symmetric tensors and specific tensors called swap-gates, which produce the correct sign changes [61, 62]. An alternative approach uses Grassman tensors [63,64]. In the future, an automated way to easily simulate fermionic systems will be implemented.

Currently, Cytnx supports only Abelian symmetries. With the current framework, it is easy to extend Cytnx to include symmetric UniTensors with more complicated symmetries such as SU(2). This is planned for a future version.

Regarding applications, several high-level tensor network algorithms such as (i)DMRG and iTEBD are implemented with Cytnx already [30]. We plan to extend this to provide a higher level abstraction in a spirit similar to the famous machine learning library Keras [65] so that physicists can focus on solving interesting physics problems. Through this, we hope to reduce the cognitive load for users and help develop more applications of tensor network algorithms in different fields of physics.

## 13  Acknowledgements

We would like to thank the National Center for High-Performance Computing (NCHC) and NVIDIA for providing us access to the NVIDIA A100 GPU. C.-M.C. acknowledges the support by the National Science and Technology Council (NSTC) under Grant No. 111-2112-M-110-006-MY3, and by the Yushan Young Scholar Program under the Ministry of Education (MOE) in Taiwan. We thank Yu-Cheng Lin and Shih-Hao Hung for their help on the implementation of the GPU functionalities of Cytnx.

## A  Tensor

A tensor is typically represented in Cytnx by a `UniTensor` object, which is explained in Sec. 5. In addition to the data of the tensor elements, a `UniTensor` object also contains meta information for bonds, labels, name, and symmetries. Internally, the data of a `UniTensor` is represented by one or more blocks. Each block contains a `Tensor` object, which is is a multi-dimensional array containing the values of the tensor. All elements belong to the same datatype. The behavior and the API of `Tensor` in Cytnx are very similar to the `tensor` in PyTorch and the `array` in NumPy. Typically, users do not have to deal with `Tensor` objects and we recommend the users to use `UniTensor` instead. However, we explain the underlying structure of a `Tensor` for advanced users here. A more extensive description can be found in the User Guide [30].

**Creating a Tensor.**  One can create a `Tensor` through generators such as `zero`, `arange`, `ones`:

```
# rank-3 Tensor of shape (3,4,5) with all elements set to zero.
A = cytnx.zeros([3,4,5])
# rank-1 Tensor containing the numbers [0,10) with step size 1
```

```
B = cytnx.arange(10)
# rank-3 Tensor of shape (3,4,5) with all elements set to one.
C = cytnx.ones([3,4,5])
```

Listing 65: Create a `Tensor` by generators.

Moreover, one can create a randomly initialized `Tensor`:

```
# Tensor of shape (3,4,5)
# All elements are distributed according to a normal distribution
# around 0 with standard deviation 1
A = cytnx.random.normal([3,4,5], mean=0., std=1.)
# Tensor of shape (3,4,5)
# All elements are distributed uniformly between -1 and 1
B = cytnx.random.uniform([3,4,5], low=-1., high=1.)
```

Listing 66: Create a randomly initialized `Tensor`.

**Data types.** The default data type of the elements in a `Tensor` is `double` in C++ and `float` in Python. One can create a `Tensor` with a different data type by providing an additional argument in the initialization function. It is possible to convert a `Tensor` of one data type to a different data type via `Tensor.astype`. The following example shows how to create a `Tensor` with `dtype=Type.Int64` and convert the data type to `Type.Double`:

```
# Create a Tensor of 64bit integers
A = cytnx.ones([2,2], dtype=cytnx.Type.Int64)
# Change the dtype to Double
B = A.astype(cytnx.Type.Double)
# Print and check
print(A.dtype_str()) # Output: Int64
print(B.dtype_str()) # Output: Double (Float64)
```

Listing 67: Data type of a `Tensor` and data type conversion.

**Reshaping.** The indices of a tensor can be combined or split, or brought in any other form of indices that match with the number of elements of the tensor. For this, the method `reshape` can be used on a `Tensor`. Suppose we want to create a rank-3 `Tensor` with `shape=(2,3,4)`. We can start with a rank-1 `Tensor` with `shape=(24)` initialized using `arange`, and then reshape it to the desired shape. This can be done by using the `Tensor.reshape` function. Note that calling `Tensor.reshape` returns a new `Tensor` object, and the shape of the original `Tensor` object does not change. If one wants to change the shape of the original `Tensor`, one should use `Tensor.reshape_`.

```
# Create a Tensor with shape=(24), initialized with the numbers
# 0,1,2,...,23
A = cytnx.arange(24)
# Reshape the Tensor to shape=(2,3,4)
B = A.reshape(2,3,4)
# Print and check
print(A.shape()) # Output: [24]
print(B.shape()) # Output: [2, 3, 4]
```

Listing 68: Reshape a `Tensor`.

**Permutation.** If the order of indices shall be changed, one can call `permute`. Consider a rank-3 `Tensor` with `shape=(2,3,4)` as an example. This time we want to permute the order of the `Tensor` indices from `(0,1,2)` to `(1,2,0)`. This can be achieved with `Tensor.permute`:

```
# Create a Tensor with shape (2,3,4)
A = cytnx.arange(24).reshape(2,3,4)
# Permute the indices such that the new shape is (3,4,2)
B = A.permute(1,2,0)
# Print and Check
print(A.shape()) # Output: [2, 3, 4]
print(B.shape()) # Output: [3, 4, 2]
```

Listing 69: Permute the indices of a `Tensor`.

**Converting to/from a NumPy array.** A `Tensor` can be converted to an NumPy array, (see also Sec. 5.8).

```
import numpy as np
# Create a Tensor
T = cytnx.ones([3,4])
# Convert to NumPy
nT = T.numpy()
print(type(nT)) # Output: <class 'numpy.ndarray'>
```

Listing 70: Convert a `Tensor` to NumPy.

A conversion from NumPy to Cytnx `Tensor` is possible similarly:

```
import numpy as np
# Create a NumPy array
nT = np.ones([3,4])
# Convert to Cytnx
T = cytnx.from_numpy(nT)
print(type(T))  # Output: <class 'cytnx.cytnx.Tensor'>
```

Listing 71: Convert a NumPy array to a `Tensor`.

**Linear Algebra.** Cytnx provides several tensor operations and linear algebra functions like SVD, QR- and eigenvalue decomposition. These are explained in Sec. 9 for `UniTensor` objects, but can be used in a similar way for a `Tensor`. However, since a `Tensor` has no attribute `rowrank`, one has to permute and reshape the tensor to bring it into the form of a matrix before calling linear algebra functions that expect matrices as input. See the User Guide [39] for a list of supported operations and more details.

**Device.** As explained in Sec. 10, tensors can be stored either in the CPU or GPU memory. If tensors are located on the GPU, one can directly make use of the parallelization and speed up the code execution. The APIs of tensor operations and linear algebra functions work the same way independently of where the tensors are stored. The code examples in listings 72 and 73 demonstrate how to initialize a `Tensor` on a specific device and how to move a `Tensor` between devices.

```
# Create a Tensor on CPU initialized with zeros
A = cytnx.zeros([3,4,5], device=cytnx.Device.cpu)
# Create a tensor on GPU  initialized with zeros
B = cytnx.zeros([3,4,5], device=cytnx.Device.cuda+0)
# Check device
print(A) # on CPU
```

```
print(B) # on GPU
```

Listing 72: Create a `Tensor` on a CPU or GPU.

```
A = cytnx.ones([2,2])
B = A.to(cytnx.Device.cuda+0)
print(A) # on CPU
print(B) # on GPU
A.to_(cytnx.Device.cuda)
print(A) # on GPU
```

Listing 73: Move a `Tensor` between CPU and GPU.

# B  Advanced usage of Cytnx

**Contiguous.**   Here, we explain a more subtle behavior in permuting index order. In Cytnx, the `permute` operation does not move the elements in the memory immediately. Only the meta-data is changed. Permuting the index order means relocating the data in memory, which requires copying all the data in a `Tensor` or `UniTensor`. This can be expensive when the tensor dimensions are large. Instead, we follow an approach that is commonly used, for example in `NumPy.array` and `torch.tensor`: When `permute` is used, the order of the data is *not* rearranged yet. The actual copying of data to the new layout in memory only happens when `Tensor.contiguous` or `UniTensor.contiguous` is called. With this design, one can permute the indices several times without significant costs in execution time, and call `contiguous` only before an actual calculation is used which depends on the order of the elements – for example, before a tensor contraction. In practice, users do not need to call the `contiguous` command by themselves, as it is automatically done by Cytnx when needed.

We can use `UniTensor.is_contiguous` or `Tensor.is_contiguous` to check if a tensor is in contiguous form. One can force the contiguous status by calling `Tensor.contiguous` or `Tensor.contiguous_` (similarly for `UniTensor`). However, the user does typically not need to do this explicitly as Cytnx automatically brings tensors to contiguous form when necessary.

```
# Create a Tensor with shape (2,3,4)
A = cytnx.arange(24).reshape(2,3,4)
# Check if it is contiguous
print(A.is_contiguous()) # Output: True
# Permute the tensor indices to a new shape (3,2,4)
A.permute_(1,0,2)
# Check if it is contiguous
print(A.is_contiguous()) # Output: False
# Make the tensor contiguous
A.contiguous_()
# Check if it is contiguous
print(A.is_contiguous()) # Output: True
```

Listing 74: Check contiguous form and make contigous.

**Storage.**   `Storage` is the low-level container that handles the memory allocation and the transfer between different devices. The tensor elements of a `Tensor` object are stored inside a `Storage` object. Typically, users do not directly need to access this object, but it might be useful in special cases. A `Storage` object can be created in a similar way as a `Tensor`. Note that `Storage` does not have the concept of `shape`, and is similar to the `std::vector` in C++. The code below in Listing 75 shows how to create a `Storage` with all elements set to zero,

and how to change the data type. One can call `Storage.astype` to convert between different data types, similar to the conversion of a `Tensor`. For details and available data types, see Appendix A.

```
# Create a Storage with 10 elements with data type Double
A = cytnx.Storage(10, dtype=cytnx.Type.Double)
# Set all the elements to zero
A.set_zeros()
# Change the data type to ComplexDouble
B = A.astype(cytnx.Type.ComplexDouble)
# Print the Storage
print(A)  # Part of output: dtype : Double (Float64)
print(B)  # Part of output: dtype : Complex Double (Complex Float64)
```

Listing 75: Create and print a `Storage`.

Internally, the data of a `Tensor` is stored in a `Storage` object. We can get the `Storage` of a `Tensor` using `Tensor.storage`:

```
# Create a Tensor
A = cytnx.arange(10).reshape(2,5)
# Get its Storage
B = A.storage()
# Check objects
print(A)  # Part of output: dtype : Double (Float64)
print(B)  # Part of output: size  : 10
```

Listing 76: Get `Storage` from a `Tensor`.

When a `Tensor` is non-contiguous (see Appendix B), its meta-data is detached from its memory. The latter is handled by a `Storage`. Consequently, calling `Tensor.storage` returns its current memory layout, not the permuted order of the `Tensor`. The following code example demonstrates this behavior:

```
# Create a Tensor
A = cytnx.arange(8).reshape(2,2,2)
# Print its Storage
print(A.storage())      # Storage elements: [ 0 1 2 3 4 5 6 7 ]
# Make it non-contiguous
A.permute_(0,2,1)
print(A.is_contiguous())  # Output: False
# Note that the storage is not changed
print(A.storage())      # Storage elements: [ 0 1 2 3 4 5 6 7 ]
# Make it contiguous, thus, the elements are permuted in memory
A.contiguous_()
print(A.is_contiguous())  # Output: True
# Note that the storage now is changed
print(A.storage())      # Storage elements: [ 0 2 1 3 4 6 5 7 ]
```

Listing 77: Reorder elements of `Storage` after `permute`.

**OpenMP.** If OpenMP is available, Cytnx uses it for the parallel execution of code on CPUs. If Cytnx is compiled without OpenMP, `Device.Ncpus` (see Sec. 10.1) is always 1. If OpenMP is enabled, one can change the environment variable `OMP_NUM_THREADS` before executing the program to set a restriction on how many threads the program can use. For example, the following code example makes OpenMP and the internal functions of Cytnx use 16 threads where possible:

```
# In the command prompt, before executing a Cytnx program:
export OMP_NUM_THREADS=16
```

Listing 78: Set the number of threads used by OpenMP.

The speedup when using OpenMP and eight threads instead of only one thread can be seen in the benchmark in Fig. 18.

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
