# Peer review of "The Cytnx Library for Tensor Networks"

_SciPost Physics Codebases, doi:SciPost Phys. Codebases 53-r1.0 (2025) , SciPost Phys. Codebases 53 (2025)_

## Round 2 · Referee Report · Matteo Rizzi (Referee 2) · 2025-2-3

Report

I am very glad to see that the Authors took the "polishing round" very seriously. The readers and users will appreciate the extra effort.
As stated about the previous version already, the Manuscript and the Code certainly fit the criteria for SciPost Physics Codebases and have the potential to belong to its higher-end.
And, as predicted, I do not see any remaining hindrance towards publication.

Recommendation

Publish (surpasses expectations and criteria for this Journal; among top 10%)

---

## Round 2 · Referee Report · Anonymous (Referee 1) · 2025-2-7

Report

I am impressed by the recent changes. I recommend publication.

Recommendation

Publish (easily meets expectations and criteria for this Journal; among top 50%)

---

## Round 2 · Author Response

Overall reply

We thank the referees for the comments. In this resubmission, we have revised the manuscript according to these comments. Before we reply to each comment in detail, we would like to reiterate that the main goal of the manuscript is to provide an overview of the functionality and usage of Cytnx. We demonstrate that the library is easy to use, versatile, and the code is readable through the use of user-defined labels and label-based network contraction. On the other hand, through benchmarking, we show that Cytnx has a similar performance as other established tensor network libraries on CPU and GPU. Consequently, uses can focus on the applications while benefiting from the speed provided by Cytnx. However, the manuscript does not aim to provide a comprehensive benchmark on tensor network algorithms against all existing tensor libraries. Neither does it aim to focus on higher-level applications, which are on our future roadmap.

Report-1

Strengths

Cytnx combines easy-to-use APIs for tensor operations with advanced performance features like memory optimization, catering to both novices and experts. It supports symmetric tensors and uses GPU acceleration, greatly improving computational speed and scalability. I would say this library is an important contribution in computational physics. The technical methodologies employed in the development of the Cytnx library appear sound. The benchmarks provided demonstrate competitive performance with iTensor.

Weaknesses

1) The documentation could benefit from including implementations of common tensor network algorithms such as DMRG (Density Matrix Renormalization Group), TDVP (Time-Dependent Variational Principle) for non-trivial problems. Their absence misses an opportunity to provide users with practical starting points for using the library in well-established computational frameworks.

We thank the referee for this comment. However, as previously stated, the goal of this manuscript is to provide an overview of the library. It does not aim to focus on high-level applications. In our future roadmap, we plan to implement additional key tensor network algorithms and provide a higher level abstraction in a spirit similar to Keras. We are currently working on a github repo https://github.com/Cytnx-dev/Cytnx_examples that provides example codes for several common algorithms.

2) Although the authors mention its potential applicability to higher dimensional tensor network techniques like PEPS, there are no explicit demonstrations that Cytnx can provide a competitive edge.

3) In general, the library seems well equipped but no clear demonstation that it can solve challenging problems. Maybe the future versions will include that.

In the revised manuscript, we add benchmark for the two-dimensional corner transfer matrix renormalziation group algorithm, which is a key ingredient in a typical iPEPS calculation for 2D quantum system. We show that the performance is similar to PEPS-Torch. This shows that Cytnx has great potential in higher dimensional tensor network algorithms.

4) It would be great if the authors could expand on the discussion of potential applications in quantum computing and machine learning to highlight the library's versatility and potential impact further.

We thank the referee for the suggestions. In the Introduction and Summary, we have added text to highlight potential applications in other fields thanks to the versatility of Cytnx. In the revised manuscript, we also included a quantum circuit simulation example in the App. C.

Report-2

Report

The manuscript presents a valuable contribution to the field of computational physics, introducing a versatile and efficient tool for tensor network simulations. With some revisions and/or comments which potentially addresses its weaknesses, it would be a suitable candidate for publication in SciPost.

Strengths

1 - accessible presentation, quick learning curve 2 - flexibility of GPU implementation at essentially zero-cost 3 - similarity to the widespread notation of other scientific libraries, hopefully easing compatibility 4 - presentation of benchmarks

Weaknesses

1 - scarce discussion of the relation to other works, especially when it comes to automatic selection of contraction order 2 - syntax is in some places slippery, with commands differing only slightly and thus being prone to generate bugs 3 - benchmarks are a bit reductive, and might be extended to other typical tensor network cases (and should be presented for both programming languages)
In the revised manuscript, we have add more discussion on related works and reworked the API. We have also added a 2D corner transfer matrix renormalization group algorithm to the benchmark.

Report

The Authors present here a tensor network library (Cytnx) developed in Python and C++, that aims at a quick learning process for new users. It resembles in most of its syntax other popular libraries, and provides a smooth integration of GPU calculations. It leverages over established formalism and concepts in tensor networks (e.g., definition of multiple Abelian symmetries), and promises to be soon extended by others (e.g., treatment of fermions and of non-Abelian groups, or support for automatic differentiation). It also integrates an automatic evaluation of the optimal contraction order for large tensor networks. Benchmarks are offered for a couple of typical operations in the most common tensor network algorithm, namely density matrix renormalization group over matrix product states.

Overall, the Manuscript presents itself as a detailed User’s guide, with precious code snippets that smoothen the learning curve, offering a quick hands-on to everyone. The overall flow of reading feels quite natural, although certain things might be defined in a different order / not anticipated so much -- but that is certainly only matter of personal taste. It falls however a bit short when it comes to put things in the broader context of other existing works (see below for some suggestions).

In general, the Library seems to allow for a lot of smart syntax to ease the life of the programmers, e.g., a flexible naming of the indices and the possibility of forgetting about their precise position for most uses. In some places, though, the command naming seems at first sight a bit slippery (see below). Some additional benchmarks are desirable, e.g., for typical contractions in two-dimensional algorithms, and for the automated selection of contraction order.

Summarising, while it certainly fits the criteria for SciPost Physics Codebases and has the potential to belong to its higher-end, the Manuscript could still be improved / enriched before publication. I do not see major hindrances that could make the process other than smooth: The work presented here is very good, and would just profit of a polishing round. Let me list some more specific points below, in the hope that the Authors would see my point and could in case spot further knobs to be tuned.

Incidentally, I sincerely apologise for the delay incurred in preparing this Report, which was not in any way related to the quality of the Manuscript.

Requested changes

1) Concerning other works that discuss the implementation of tensor network contractions and the conservation of quantum numbers, I find it remarkable that works by Montangero, Weichselbaum, and others (some of which on SciPost) are not mentioned. Same applies, even more urgently, when discussing a feature like automated decision of the optimal contraction sequence: a work by Pfeifer, Haegeman and Verstraete bearing almost this exact title should be mentioned.

We thank the referee for pointing this out. In the revised manuscript, we have added the following references in relevant sections: * [9] P. Silvi, F. Tschirsich, M. Gerster, J. Jünemann, D. Jaschke, M. Rizzi and S. Montangero, The Tensor Networks Anthology: Simulation techniques for many-body quantum lattice systems, SciPost Physics Lecture Notes p. 008 (2019), * [40] A. Weichselbaum, Non-abelian symmetries in tensor networks: A quantum symmetry space approach, Annals of Physics 327(12), 2972 (2012). * [41] A. Weichselbaum, X-symbols for non-Abelian symmetries in tensor networks, Physical Review Research 2(2), 023385 (2020). * [50] R. N. C. Pfeifer, J. Haegeman and F. Verstraete, Faster identification of optimal contraction sequences for tensor networks, Physical Review E 90(3), 033315 (2014).

2) Minor: The naming UniTensor suggests at first reading some relation to unitarity, which is not at all the case — what is the reason behind the choice? Almost surely a legacy to Uni10 library, which is however not discussed (sorry if I accidentally overlooked it).

In Sec-2 "Sneak preview", we have added text to describe the difference between UniTensor and Tensor in Cytnx. Furthermore, UNI10 is now cited as the origin of such a naming convention.

3) In Listing 2 (and elsewhere, too), why are such similar names like “Contract” and “Contracts” chosen? A typo is extremely likely to occur, causing an easily avoidable bug. Is there any control in place to prevent this to happen? (Same applies later to “.relabel_” and “.relabels_”, just to cite one). By the way, at this stage of the presentation, it is not clear how “Contracts” will decide over the contraction order.

4) The remark “One typically needs to first find an optimal contraction order before implementing the contractions” should at best be accompanied by the warning that this is (by far!) not the one exemplified in Listing 2, if all bonds have similar dimension…

We thank the referee for pointing this point. In the revised manuscript, we have merged these APIs to avoid confusion. Related paragraphs and code examples are changed accordingly (see also list of changes). We have also added some text after Listing 2 to comment on the contraction order, but we defer the full discussion about contraction orer to Sec-7 and Sec-8.

5) Speaking of slippery notation, and possible typos: is there any specific meaning attached to the “;” between the indices of TOUT in Listing 3? The indices of the other two tensors are separated by “,” instead.

We thank the referee for pointing this out. Originally, the ";" in TOUT was used to specify the row-rank of the TOUT tensor. We have changed the API of the .Network() so that TOUT will obtain a default row-rank and it is not necessary to use ";" and the indices in TOUT are always separated by ",". In the revised manuscript, all relevant code examples are modified accordingly.

6) Minor: When discussing the merging of bonds in Sec. 4, the natural question is how this will be ruled when symmetries are present. May a link to the proper place in Sec. 6 be useful here? Incidentally, is the reverse operation (i.e., splitting a bond in multiple ones) implemented, and how?

Now in Sec-4 we provide a link to Sec-6.2 (symmetric bonds). At this moment, no reverse operation is implemented.

7) Minor: A few commands seem a bit involved, like in Listing 16 the apparent need of assigning explicitly a name to a UniTensor, identical to the variable name on the lhs of the “=“. Or is there any arbitrariness that could be useful in specific cases? If so, a comment would be desirable.

We have added text to explain that the name of a UniTensor can be any string and is independent of the variable name.

8) When discussing the possible initialisations of a tensor in Sec. 5, a natural question is whether one could add a bit of noise on top of existing entries loaded from a file (e.g., tensors coming from another simulation, or in the case of single-site DMRG algorithms with so-called subspace expansion). In other words, can one perform an element-by-element sum or rescaling of a tensor? It might be related to Secs. 5.5 and 5.6, but it could be useful to mention it more clearly.

We thank the referee for the comment. In the revised manuscript, we have moved the section "Arithmetic operations and tensor operations" to be right after the section "Creating a non-symmetric UniTensor" to show how to do an element-by-element sum, or rescaling. Furthermore, we now show how to add noise to an existing UniTensor in Listing 20.

9) Minor: on page 15, there is a discussion about “rowrank” deciding over the left/right position of indices in the pictorial representation of the tensor (“We observe that bond "a" is attached to the left of the tensor, while bonds "b" and"c" are attached to the right…”) but this does not correspond to the illustration in Listing 22. Please amend.

We thank the referee for pointing this out. In the revised manuscript, we have modified Sec-5.5 to properly link the diagram in Listing 23 and the text. Furthermore, we have added examples to explain how to change the rowrank and observe how the diagram changes accordingly.

10) In Listing 24 and 25, two notations seem to be present for indicating a specific element of a UniTensor, namely “uT.at(["a","b"],[0,2]).value” and “uT[0,2]”: are they truly identical? If yes, why should one ever want to use the first? If not, where is the crucial difference?

We thank the referee for asking this question and the short answer is "yes". In order to clarify this, we have revised Listing 25 and use a rank-3 tensor as an example to illustrate how to use different syntax to achieve identical results. If a list of labels is not given, then the internal order of the indices in the UniTensor is used. In contrast to this, if a list of labels is given, then the label and index pairs are matched to identify the element. Here, the key point is that the list of labels does not need to follow the internal ordering of the indices in the Unitensor. In other words, users can specify the legs via labels without knowing their internal ordering.

11) As a practitioner of tensor networks with symmetries, I am interested in understanding why “users can choose not to group them [sectors with the same quantum number] by setting the argument “is_grp=False”. When is that useful? Can the Authors provide a concrete example for the Readership?

In the revised manuscript, we have moved this to the footnote since this is an advanced feature. The feature can be useful when one has other properties in addition to the Abelian symmetry and needs to keep track of the state manually.

12) Minor: Is there any constraint on the dimensions of incoming / outgoing bonds of symmetric tensors, or are they completely free?

There is no constraint.

13) When discussing linear algebra operations with symmetric tensors, the case of truncated SVD should be discussed in more detail. The comparison of values to be thrown away should be performed across different sectors, to exclude the risk of spurious symmetry breaking effects. This is discussed in multiple sources about DMRG and other TN algorithms, and should be recalled to the reader here, in the spirit of allowing for “quick learning process for new users”.

We thank the referee for the comment. In the revised manuscript, we have added a paragraph to discuss SVD/SVD_trunc for a symmetric UniTensor. We explain that for a symmetric UniTenor, all singular values from all the symmetry sectors are sorted together to decide how to truncate. We also show how a minimum number of singular values can be enforced in each block with an additional argument min_blockdim. Moreover, new examples are added to illustrate the behavior for symmetric UniTensors.

14) In Sec. 8 about the “Network” object, its relation to old-standing community-routines like “ncon” and alike might be worth to be discussed. Same, and even more importantly, the similarities and differences between the automated generation of contraction here and the one proposed one decade ago in works by Pfeifer et al.

In the revised manuscript, we have modified two sections to better discuss this issue: In Sec-7, we have added text to discuss the need for the automated generation of an optimal contraction order and add relevant references. We explain that all three methods (Contract, ncon, and Network) in Cytnx support this. We also add the references for our implementation.

In Sec-8, we discuss the main difference between Network and ncon: Network is label based and user can use physically meaningful strings as labels. This makes the code more human readable and easier to debug. Furthermore, users never need to worry about the internal ordering of the labels. Also, the optimal contraction order can be re-used in Network for subsequent contraction of tensors with similar dimensions.

15) In Sec. 9.2, it might be worth to mention the possibility of using Golub-Kahan-Lanczos bidiagonalization (like in KrylovKit.jl, for example) for obtaining a truncated SVD without need of computing first the full one. While this is commonly not useful in a DMRG algorithm with small physical dimension, it might become handy when the fraction of the singular values to be kept is small (e.g., in a PEPS or TTN application, and also for DMRG with large local dimension). The spirit is very similar to what the Authors hint at when discussing diagonalization of the lowest spectrum via Lanczos in Sec. 9.3.

We thank the referee for the suggestion. We have added text at the end of Sec-9.2 to discuss the possibility of SVD_trunc without full SVD.

16) A benchmark exclusively based on C++ seems a bit partial: given the popularity of Python (and Julia), it would be very informative to see the analogue of Figs. 18-19 for these languages, too. Only so could the interested reader / user take a factual decision on what to use. Incidentally, why is the benchmark performed only against iTensor and none of the other listed libraries (e.g., TensorKit or TenPy)?

19) It is desirable to see benchmarks of the different libraries for TN algorithms beyond the "simple" DMRG, especially in the direction of two-dimensional systems (PEPS, TTN, etc.). The more complicated the operations, the more the advantage of an implementation could jump to the eyes.

In the revised manuscript, we have added a benchmark of the corner transfer matrix renormalization group (CTMRG) algorithm for the contraction of a 2D tensor network in the context of iPEPS. We compare the average time for CTMRG algorithm to obtain the converged corner tensor by PEPS-Torch and the implementation in Cytnx on a CPU. We find that the performance is similar. As mentioned in the overall reply, the manuscript aims to provide an overview of Cytnx and does not aim to provide comprehensive benchmarks against all libraries.

17) Side comment on Fig. 18: actually, the plot makes it very apparent that it is usually not a good idea to use multiple threads at least for “simple” DMRG algorithms: The gain is most often far less than the number of threads, which means that it would almost only result in a higher amount of core-hours to be accounted on HPC machines, without tangible benefit.

We thank the referee for pointing this out. However, we think core-hours is not the main concern in this manuscript. Users who want to find the best utilization of a HPC machine in their case should be able to deduce the necessary information from the figures.

18) Looking at Fig. 19, could the Authors provide an estimate at which bond dimension will the memory limitations of a GPU kick in and invert the trend?

In the paragraph for GPU benchbarmk, we have added the comment that for $d=2, M_D=4, N=40$, one can run finite-size DMRG on NVIDIA A100 GPU (80G) up to $D\approx 6000$.

20) Similarly, it would be good to have concrete examples where the automatic selection of the contraction sequence does bring a concrete advantage — no doubts it is a nice feature to have, but it is good to see it in action.

We thank the referee for this comment. In light of this comment, we have also modified our code to incorporate the cost-capping depth-first algorithm described in Ref. [50], which significantly reduced computation time for obtaining the optimal contraction order.

21) Side comment: When mentioning automated differentiation among future features to be implemented, I cannot agree more, especially given the prominent role AD plays in unlocking (making more accessible and flexible) the variational optimization of complex TN structures like PEPS. Mentioning it could be helpful for interested readers.

We thank the refree for the suggestion. We changed the corresponding sentence to "Automatic differentiation has revolutionized variational optimization of complicated tensor networks such as PEPS [67–70], and provided simpler formulation to extract information about physical observables related to experiments, such as spectral functions [71]."

---

## Round 2 · List of Changes

In the following we provide a brief list of major changes and a list of API changes.

List of changes

  • In Sec-2: sneak preview:
  • We add text to briefly discuss the distinction between UniTensor and Tensor in Cytnx and cite UNI10 as the origin of this naming convention.
  • We modify the text according to the new API (.Contract) and mention the optimal contraction order.
  • In Sec-5.1: Creating a non-symmetric UniTensor
  • We add text to show how to create a Tensor and then convert it to a UniTensor.
  • We add text to clarify that the name of a UniTensor can be any string.
  • We explain the seed number argument for randomized tensors to make our code examples better reproducible
  • In Sec-5.2: Arithmetic operations and tensor operations
  • We add text to show how to add random noise to a UniTensor.
  • In Sec-5.3: Viewing/Changing labels
  • We modify the text according to the new API (.relabel).
  • In Sec-5.5 Tensor information
  • We modify the text to be consistent with the Listing 23 and add text to clarify the meaning of rowrank.
  • In Sec-5.6 Getting/Setting elements
  • We modify the text to better explain how to use labels to get/set elements and its advantage. We also add comments about symmetric UniTensors.
  • In Sec-6.2.2 Combining bonds with quantum numbers
  • We clarify that only bonds with the same type and symmetries can be combined.
  • We move the "not grouping" option is_grp=False to the footnote.
  • In Sec-6.3.1 Creating a symmetric UniTensor
  • We add text to address the arithmetic of symmetric UniTensors.
  • In Sec-7 Tensor contraction
  • We re-write the section to better explain Cytnx's support for Contract and ncon. We add text to better explain the automatically generated optimal contraction order.
  • In Sec-8 Network
  • We add text to emphasize the advantage/difference of Network over ncon.
  • In Sec-9.2 Singular value decomposition
  • We add text to better explain the SVD for symmetric UniTensors, especially how Svd_truncate works. We add comments about potential future implementations of the truncated SVD without a full SVD.
  • In Sec-11 Benchmarks
  • We add a new benchmark: CTMRG algorithm for the contraction of a 2D tensor network.
  • In App-C Quantum Circuit Simulation
  • We add a section describing how to use Cytnx to simulate quantum circuits.

List of API changes

  • Contract, Contracts are merged into Contract
  • .relabel, .relabels are merged into .relabel
  • .relabel_, .relabels_ are merged into .relabel_
  • Svd_truncate has an additional optional argument min_blockdim to define a minimum dimension for each block
  • Code examples are modified accordingly.

---

## Editorial Decision

published